# Integrative profiling of condensation-prone RNAs during early development

## Graphical abstract

## Authors

Tajda Klobučar, Jona Novljan, Ira A. Iosub, ..., Nicholas M. Luscombe, Jernej Ule, Miha Modic

## Correspondence

miha.modic@kit.edu

## In brief

This study identifies a unique class of condensation-prone RNAs (smOOPs), defined by semi-extractability and enrichment in OOPS. smOOPs form denser-than-expected RNA-RNA subnetworks, display distinct sequence features, and are strongly bound by RBPs. They encode proteins rich in intrinsically disordered regions, suggesting a coordinated RNA-protein interplay in phase separation.

## Highlights

- Semi-extractability and OOPS uncover developmental condensation-prone RNAs (smOOPs)

- smOOPs are in close proximity, reflected by their denser-than-expected RNA subnetworks

- Multi-omic datasets link transcript features to condensation potential

- smOOPs encode proteins rich in intrinsically disordered regions prone to phase separation

 Klobučar et al., 2026, Cell Genomics 6, 101065
February 11, 2026 © 2025 The Author(s). Published by Elsevier Inc.

CellPress

## Article

# Integrative profiling of condensation-prone RNAs during early development

Tajda Klobučar,[1,2,8,12] Jona Novljan,[1,3,4,5,12] Ira A. Iosub,[2,6,7,12] Boštjan Kokot,[9] Iztok Urbančič,[9] D. Marc Jones,[2,6,7] Anob M. Chakrabarti,[2,10] Nicholas M. Luscombe,[2,11] Jernej Ule,[1,2,6,7] and Miha Modic[1,2,3,4,5,6,13,*]

[1]National Institute of Chemistry, Ljubljana, Slovenia
[2]The Francis Crick Institute, London, UK
[3]Department of Genomics and Developmental Biology, Zoological Institute, Karlsruhe Institute of Technology, Karlsruhe, Germany
[4]Institute of Biological and Chemical Systems, Karlsruhe Institute of Technology, Karlsruhe, Germany
[5]Center for Synthetic Genomics (SynGen) Heidelberg-Karlsruhe-Mainz, Germany
[6]Department of Basic and Clinical Neuroscience, Institute of Psychiatry, Psychology and Neuroscience, King's College London, London, UK
[7]Dementia Research Institute at KCL, London, UK
[8]PhD Program "Biosciences", Biotechnical Faculty, University of Ljubljana, Ljubljana, Slovenia
[9]J. Stefan Institute, Ljubljana, Slovenia
[10]University College London, UCL Respiratory, London, UK
[11]Okinawa Institute of Science and Technology, Okinawa, Japan
[12]These authors contributed equally
[13]Lead contact
*Correspondence: miha.modic@kit.edu

## SUMMARY

Complex RNA-protein networks play a pivotal role in the formation of many types of biomolecular condensates. How RNA features contribute to condensate formation, however, remains incompletely understood. Here, we integrate tailored transcriptomics assays to identify a distinct class of developmental condensation-prone RNAs termed "smOOPs" (semi-extractable, orthogonal-organic-phase-separation-enriched RNAs). These transcripts localize to larger intracellular foci, form denser RNA subnetworks than expected, and are heavily bound by RNA-binding proteins (RBPs). Using an explainable deep learning framework, we reveal that smOOPs harbor characteristic sequence composition, with lower sequence complexity, increased intramolecular folding, and specific RBP-binding patterns. Intriguingly, these RNAs encode proteins bearing extensive intrinsically disordered regions and are highly predicted to be involved in biomolecular condensates, indicating an interplay between RNA- and protein-based features in phase separation. This work advances our understanding of condensation-prone RNAs and provides a versatile resource to further investigate RNA-driven condensation principles.

## INTRODUCTION

Cells exhibit a wide range of RNA assemblies that physically partition into subcellular membraneless compartments or biomolecular condensates, but the general molecular rules governing RNA condensation and their local entrapment in ribonucleoproteins (RNPs) remain unclear.[1,2] RNA-binding proteins (RBPs) and RNAs have both been implicated in condensate formation, and disruptions in their phase separation have been linked to pathological conditions, including impaired embryonic development, cancers, neurodegenerative diseases, and others.[3–5] Many proteins within RNP condensates contain intrinsically disordered regions (IDRs) that are able to form weak multivalent interactions,[6,7] and simple changes to protein sequence or charge alone can drastically alter their condensation properties.[8,9] Conversely, RNA molecules can drive condensation themselves, either as a scaffold or through RNA-RNA interactions (RRIs).[10–15] Recently, G3BP1 was shown to work as an

"RNA condenser" that promotes intermolecular RRIs that stabilize stress granules,[16,17] while exceptionally long cytoplasmic mRNAs were shown to scaffold FXR1 protein into a network-mediating signaling response.[18] Understanding which and how RNA features contribute to condensate formation and function, particularly through their interplay with RBPs, remains a challenging question.

Several transcriptomic approaches opened the avenue for exploring condensation principles in an RNA-centric manner. Some of these methods are capturing RNAs based on their biochemical properties and their associations with RBPs, which are key for recruitment into and stabilization within condensates.[19] Studies on semi-extractable RNAs[20,21] identified a diverse array of RNA species associated with biomolecular condensates, particularly within nuclear bodies.[21] UV-crosslinking-based methods recover RBP-bound RNAs.[21–23] In addition, RNA proximity-ligation approaches[24] furthered our understanding of higher-order RNA structures, e.g., in stress



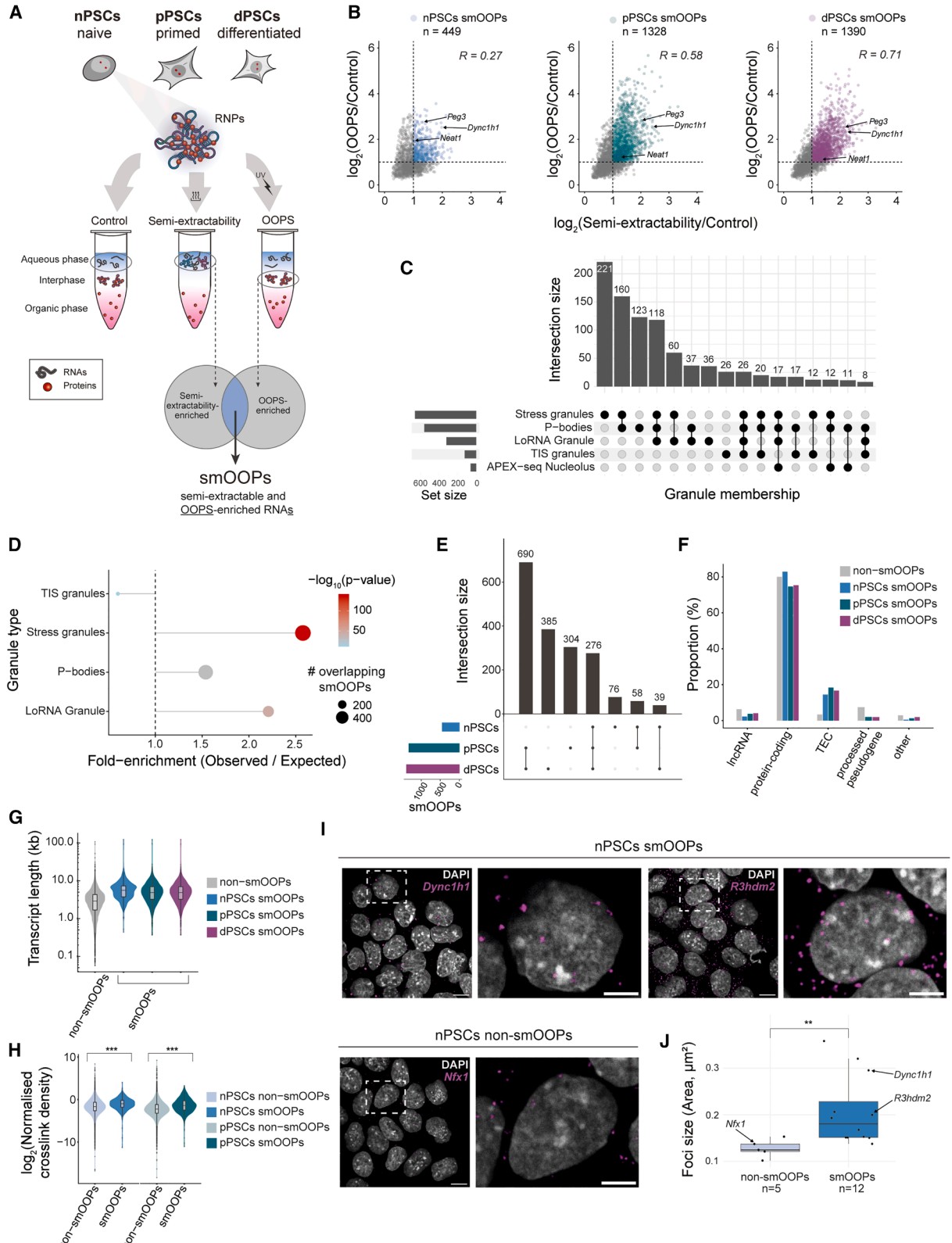

*(legend on next page)*

granules,[25] as well as intermolecular RRIs, such as those between enhancer RNAs/mRNAs,[26] small nucleolar RNAs (snoRNAs)/target RNAs,[25] and viral/host transcripts.[27]

To advance our current understanding of RNA-centric condensation features, we designated a novel class of transcripts as "smOOPs" (semi-extractable, orthogonal-organic-phase-separation-enriched RNAs) due to their semi-extractability[20] and pronounced affinity for orthogonal organic phase separation (OOPS).[22] Together, these two methods provide a comprehensive strategy to identify candidate RNAs that may be involved in condensation processes—that is, condensation-prone RNAs, which we define as highly interacting RNA molecules that are likely to enrich and concentrate in phase-separated compartments or other RNP assemblies. This group includes RNAs known to form or localize within condensates, alongside other RNAs sharing similar properties, which we hypothesize are prone to condensation. Through RNA *in situ* conformation sequencing (RIC-seq),[26] we show that smOOPs are in greater proximity to one another, suggesting that these RNAs are closely associated within cells. Using a combinatorial deep learning (DL) approach, we identified core smOOPs features and revealed that these transcripts code for highly disordered proteins with elevated phase-separation propensity. Notably, this study offers a comprehensive methodological framework for uncovering the principles of RNA assembly and their possible role in coordinating post-transcriptional gene regulation, accelerating the extraction of condensation-relevant features from diverse datasets.

## RESULTS

### A class of semi-extractable and OOPS-enriched RNAs across early development

To specifically enrich RNA molecules within RNP assemblies, we employed a combination of the semi-extractability assay[20] and OOPS[22] (Figure 1A). We applied these methods to recover such RNAs during three distinct time points of early embryonic development: naive pluripotent stem cells (nPSCs) in 2iLif condi-

tions[28]; primed epiblast stem cells (pPSCs), in which lineage priming was prevented by Wnt inhibition[29,30]; and the earliest Wnt-differentiated primitive streak progenitors (dPSCs)[31] (see STAR Methods). A standard TRIzol RNA extraction protocol was used as a control. We generated total RNA sequencing (RNA-seq) libraries that showed high correlations in gene-level counts within each developmental stage and assay type (Figure S1A). In a principal-component analysis (PCA), the samples separated primarily by developmental stage (76% of the variance), with nPSCs being more distinct from pPSCs and dPSCs. The assay type further contributed to the separation (14% of the variance), with the OOPS samples being the most distinct (Figure S1B).

We next performed differential expression analysis to identify semi-extractable and OOPS-enriched genes compared to standard TRIzol RNA-seq controls at each developmental stage (Figures S1C and 1B; Data S1, S2, S3, S4, S5, and S6). By employing stringent effect-size and statistical cutoffs (at least 2-fold enrichment and an adjusted p value [padj] < 0.01), we identified 449, 1,328, and 1,390 high-confidence genes at each developmental stage with distinctly increased semi-extractability and elevated RBP occupancy in OOPS samples, henceforth collectively referred to as smOOPs (Figures 1B–1E and S1C; Table S1). Among smOOPs, we recovered RNAs known to form condensates, such as *Neat1*, involved in paraspeckles[20,32]; *Dync1h1*, known to form cytoplasmic foci of 3–7 copies at active translation sites in *Drosophila*[33]; and *Peg3*, for which the human homolog was found to be enriched in stress granules.[34] These examples underscore the inclusion of known condensate-forming RNAs within the smOOPs group (Figure 1B). To assess this more systematically, we intersected smOOPs with high-throughput datasets of condensate-enriched RNAs (derived from human cells using ortholog gene transfer to mouse[35–39]). We observed that smOOPs do not belong to a single granule type but rather are heterogeneously distributed across multiple condensate classes (Figure 1C). Notably, we found significant overlap and overrepresentation of RNAs localized to stress granules,[36] processing bodies (P-bodies),[35] and

---

**Figure 1. Atlas of semi-extractable and OOPS-enriched RNAs across early development**

(A) Experimental framework to identify RNAs that are both semi-extractable and highly RBP-bound RNAs (smOOPs) using three different TRIzol-based RNA extractions (done in three replicates): aqueous phase of non-crosslinked sample as control, aqueous phase of heated and sheared TRIzol sample as semi-extractable RNAs, and interphase of crosslinked sample to obtain OOPS-enriched RNAs (nPSCs, naive pluripotent embryonic stem cells; pPSCs, primed pluripotent stem cells; dPSCs, 1 day Wnt-differentiated pPSCs).

(B) Scatterplots showing the overlap between semi-extractability and OOPS-enriched genes (padj < 0.01) as well as the Pearson correlation between their fold changes compared to control at each developmental stage. At each stage, the genes enriched more than 2-fold compared to the control (padj < 0.01 and LFC > 1) in both assays were defined as smOOPs.

(C) Mapping of the combined set of smOOPs from nPSCs, pPSCs, and dPSCs onto known RNP granules/membraneless compartments.

(D) Enrichment of RNAs within characterized RNP granules in the smOOPs set. Overrepresentation was assessed using a hypergeometric test, restricted to granules from (C), where at least 10 smOOPs are specific to that granule.

(E) UpSet plot showing the intersections between smOOPs identified in nPSCs, pPSCs, and dPSCs.

(F) Percentage of gene biotypes for smOOPs and non-smOOPs in each cell state (TEC, to be experimentally confirmed).

(G) Comparison of transcript length distribution between smOOPs and non-smOOPs.

(H) Global iCLIP crosslinking signal normalized to expression and length and expression (crosslinks per million [CPM] density/semi-extractability TPM; pooled from three replicates) for smOOPs compared to non-smOOPs. Statistical significance was determined using a one-sided Wilcoxon rank-sum test (***p < 0.0001).

(I) Representative HCR-FISH photomicrographs (scale bar: 5 μm), with the right image showing a magnified view of the region outlined by the dotted white box (scale bar: 10 μm).

(J) HCR-FISH quantifications. The boxplot shows the mean of foci size for each target transcript, calculated as the mean of all foci for each transcript. Statistical significance was determined using a two-sided Welch's t test (**p < 0.01). n indicates the number of different mRNAs against which the HCR-FISH probes were designed. In total, >70 nuclei for a single transcript were imaged, and >890 foci were counted for each transcript.

granules obtained by the localization of RNA (LoRNA)[39] method(Figures 1C and 1D). Although based on human datasets, the observed enrichments reinforce a broader link between smOOPs and condensation propensity and indicate that they share conserved features with RNAs found in biomolecular condensates.

Of the total 1,828 unique smOOPs, 276 were common to all stages, with most occurring at later stages. Notably, there were fewer unique smOOPs identified in nPSCs (76) compared to the other stages (pPSCs, 304; dPSCs, 385; Figure 1E). A positive correlation was observed between fold changes in the OOPS and semi-extractability assays, particularly upon the onset of cell fate commitment. However, the degree of enrichment (fold change) for a gene in one assay (OOPS or semi-extractability) did not always reflect the degree of enrichment in the other (Figure 1B). Some smOOPs highly enriched in OOPS were not similarly enriched in semi-extractability (Figure S1D), indicating that each assay preferentially captures distinct RNA characteristics and that the smOOPs pool is re-wired during developmental transitions (Figures 1B–1E and S1D). Classifying smOOPs by gene biotype revealed that the majority consist of protein-coding genes (74.6%–82.2%), followed by TEC genes ("to be experimentally confirmed" genes; 14.5%–16.7%) and long non-coding RNAs (lncRNAs) (2.2%–4.1%) (Figure 1F). Notably, the TEC proportion was more than double that of the genes not classified as smOOPs (non-smOOPs) (6.3%). To better understand the potential function of TECs within smOOPs, we assessed their coding potential. Using lncRNAnet,[40] most TECs were predicted to be non-coding (Figure S1E). Furthermore, ribosome profiling data in mouse embryonic stem cells (mESCs)[41] showed no detectable translated open reading frames (ORFs) in TECs, supporting their classification as non-coding transcripts (Figure S1F). The overrepresentation of TECs in the smOOPs group suggests that these understudied transcripts might play previously unrecognized roles in RNA-centered processes independent of translation. In conclusion, our approach identifies smOOPs as transcripts that are both highly bound by RBPs and display distinct extractability, revealing a developmentally regulated class of RNAs with properties consistent with condensation propensity and RNA-protein organization.

### smOOPs: Condensation-prone RNAs form RNP granules

Our dual approach enabled us to identify smOOPs as candidate RNAs with potential for involvement in condensation processes. We observed that smOOPs are longer RNAs compared to non-smOOPs (Figure 1G), which may partially contribute to their distinct properties. To validate the tendency of smOOPs for RNP interactions, we performed global individual-nucleotide resolution crosslinking and immunoprecipitation (iCLIP)—an orthogonal method for mapping the cumulative RBP occupancy across the transcriptome.[42] This not only confirmed the elevated RBP binding compared to non-smOOPs (normalized for expression and length; nPSCs $p = 1.28 \times 10^{-15}$, pPSCs $p = 1.41 \times 10^{-49}$) (Figure 1H) but also provided precise positional information on RBP interactions (Figure S1K). We hypothesized that higher RBP occupancy, in addition to the semi-extractability of smOOPs, could indicate that they are part of RNP assemblies.

To test this, we performed hybridization chain reaction fluorescence in situ hybridization (HCR-FISH)[43] using probes against the exons of 17 candidate protein-coding transcripts, including smOOPs and non-smOOPs (Figure S1G) with largely comparable expression levels (median transcripts per million [TPM] of 21.9 for non-smOOPs and 32.5 for smOOPs; Figure S1J). Image analysis confirmed that smOOPs formed larger foci compared to non-smOOPs (Figures 1I, 1J, and S1H; Table S2), with higher overall intensity (Figure S1I), suggesting an enrichment of these RNAs in localized regions within the cell, potentially reflecting high local RNA concentrations. While these are clearly visible foci of a single RNA, this observation does not exclude the possibility that they are part of heterotypic assemblies occupying a larger subcellular area. Notably, this pattern was observed for smOOPs mRNAs that were already implicated in condensate formation: Dync1h1 and Peg3.[33,34] Further analysis of RNA distribution found that only 36.9% of tested smOOPs foci were nuclear, compared to 46.5% for non-smOOPs transcripts (Figure S1I). Taken together, our findings suggest that smOOPs are a unique class of semi-extractable transcripts that are highly bound by RBPs and form larger foci within cells. These characteristics provide evidence of their condensation-prone nature.

### smOOPs establish RNA subnetworks with enhanced connectivity

Given the well-established role of RBPs in regulating RNA assembly, we hypothesized that smOOPs participate in broader RNA networks. We applied RIC-seq[26] to map intra- and intermolecular RNA proximities across the transcriptome, capturing both direct and indirect RBP-associated RRIs in nPSCs and pPSCs (Figure 2A). We detected 758,135 hybrid reads in nPSCs and 1,245,548 in pPSCs (excluding rRNA, tRNA, and mitochondrial reads), with 28% being intermolecular in nPSCs and 44% in pPSCs (Figure 2B). Gene-level intermolecular hybrid frequency was highly reproducible across samples (Figure S2A), with PCA showing that the developmental stage explained 88% of the variance (Figure S2B). The global distribution across transcript regions remained consistent across stages, with most hybrid reads containing intronic regions (~66%, Figure S2C), similar to previous findings.[26]

We generated developmental-stage-specific RRI networks using the intermolecular hybrid reads from the RIC-seq data, with genes as nodes and connecting them with edges weighted by the frequency of hybrid reads spanning each gene pair. The inferred networks showed typical characteristics of biological networks, such as protein-protein interaction (PPI) and RRI networks.[44] Specifically, the networks displayed scale-free-like behavior, with most nodes having few connections and a small number of highly connected nodes that dominate (Figure S2D). This suggests that a few genes play central roles in the network, while most genes have fewer connections, creating a heavy-tailed distribution of connectivity (Figure S2D). The RIC-seq networks also have small-world properties, with a higher global clustering coefficient than random networks of the same size, suggesting modular organization (Figure S2E), and a relatively short average path length of 3.5, indicating that most RNAs are within short network distances, consistent with widespread RNP-mediated proximity (Figure S2F).

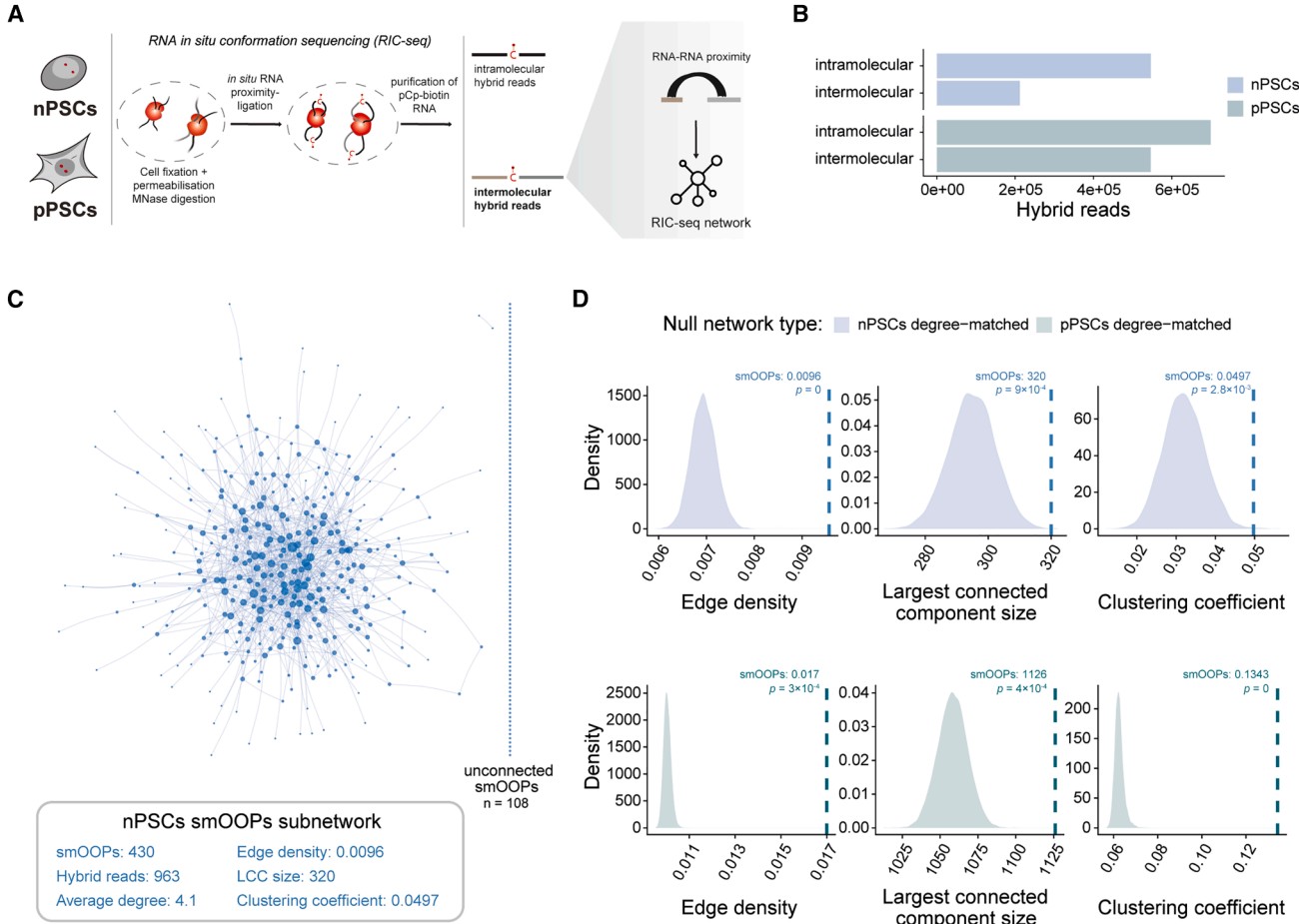

**Figure 2. smOOPs connectivity within RIC-seq networks**

(A) Schematic overview of the approach for inferring RNA-RNA networks in nPSCs and pPSCs using RIC-seq,[26] with key steps shown.

(B) RIC-seq hybrid read counts in nPSCs and pPSCs (pooled from three replicates each), categorized by type, excluding hybrid reads containing rRNA, tRNA, and mitochondrial RNA.

(C) Visualization of the nPSC smOOPs RRI subnetwork from RIC-seq data, where nodes represent genes and edges connect gene pairs supported by hybrid reads. Node size corresponds to degree, and edge width represents the number of hybrid reads between nodes. Unconnected smOOPs are displayed on the right and connectivity metrics underneath.

(D) smOOPs subnetwork connectivity comparison with degree-matched control subnetworks in nPSCs and pPSCs. Density plots show the distribution of connectivity metrics from 10,000 degree-matched sub-sampled networks (null models). The dashed lines indicate observed values for the smOOPs RIC-seq subnetworks, with $p$ values from permutation tests comparing smOOPs subnetworks' metrics to the metric distributions for the degree-matched sub-sampled networks.

Next, we explored the characteristics and connectivity of smOOPs in these RRI networks. In the nPSCs network, 430 of 449 smOOPs were present, and in pPSCs, 1,255 out of 1,328 smOOPs were present. In the RIC-seq networks, degree (the count of distinct nodes linked to each gene) strongly correlated with expression for genes of similar length, regardless of whether they were classified as smOOPs (Figure S2G), likely reflecting the higher sensitivity of RIC-seq for detecting interactions involving abundant RNAs. Although smOOPs appeared to have a high degree compared to all non-smOOPs, non-smOOPs matched for expression levels and lengths (Figure S2H) exhibited similarly high degrees (Figure S2I). This suggests that the observed elevated degree of smOOPs is primarily driven by their expression and length rather than their smOOPs status.

To explore network connectivity patterns among smOOPs, we focused on the smOOPs subnetworks within the RIC-seq data (Figures 2C and 2D). The nPSC and pPSC smOOPs subnetworks appeared highly connected based on several network connectivity metrics: edge density (the proportion of possible edges present), largest connected component (the size of the largest connected subnetwork), and global clustering coefficient (the tendency of nodes to form tightly connected groups) (Figures 2C and 2D). Because nodes with high degrees are inherently more likely to connect, we specifically tested whether smOOPs preferentially connect with each other rather than being broadly or randomly connected across the transcriptome. To do this, we generated degree-matched random subnetworks by sampling RNAs with a similar degree as smOOPs. These

degree-matched random subnetworks provide a null expectation for connectivity, enabling comparison to the observed smOOPs subnetwork while controlling for biases related to degree, expression, and transcript length. Compared to these random expectations, smOOPs subnetworks exhibited significantly greater connectivity across all metrics in nPSCs and pPSCs (Figure 2D), indicating that their interconnectivity cannot be explained by their degree alone. Furthermore, despite the identity of smOOPs varying across development (Figure 1C), this characteristic is maintained in both nPSCs and pPSCs (Figure 2D). Together, these findings suggest that smOOPs are more interconnected among themselves even when controlling for general connectivity with the whole transcriptome, reflecting a specific network organization that points to their proximity in cells.

## DL accurately predicts smOOPs from intrinsic and regulatory RNA features

Given that smOOPs are enriched in the semi-extractability and OOPS assays (Figure 1) and form denser RNA subnetworks than expected by chance (Figure 2), we pursued an in-depth investigation of the RNA features that define this group of condensation-prone transcripts. Using intrinsic RNA features—such as sequence and structure—and transcriptome-wide data for *trans*-acting factors, we developed an explainable DL approach to distinguish smOOPs from a background RNA population that are neither semi-extractable nor OOPS enriched. We focused on nPSCs due to the greater availability of public transcriptomic data compared to pPSCs, allowing us to utilize a more extensive set of features. Thus, for our binary classification, smOOPs from nPSCs with processed transcript lengths under 20 kb were used as the positive class ($n = 447$ out of 449), while genes without strong evidence of enrichment at any stage of the semi-extractable assay or OOPS vs. control ($p$adj > 0.01 and |log fold change| [|LFC|] < 1.4; see STAR Methods) were defined as the all-stage control genes ($n = 1,232$), referred to herein as common control genes (Table S1). The control genes have similar expression levels to smOOPs (Figure S3A), which strengthens our comparison by reducing potential confounding effects from expression differences. To identify the unique features of smOOPs, we trained DL classifiers with multiple feature sets: RNA nucleotide sequence, global RBP occupancy (global iCLIP; this study), N6-methyladenosine ($m^6A$) modification sites ($m^6A$-iCLIP),[45] transcriptome-wide base-pairing intramolecular (PARIS-Intra) and intermolecular (PARIS-Inter) interactions,[46] *in silico* structure prediction with RNAfold,[47] and RNA-binding sites of 46 RBPs determined via CLIP from various mouse cell lines from the POSTAR3 database.[48] We encoded all features as positional information layers for each transcript, enabling efficient feature extraction (Figure 3A; see STAR Methods). To capture complex patterns and positional dependencies in the data, we implemented a DL architecture consisting of multiple convolutional neural network (CNN) blocks, a recurrent neural network (RNN), and a multi-layer perceptron (MLP) for classification (Figure 3A). The main goal was not merely to achieve accurate predictions but also to extract and understand the discriminating power contributed by each dataset, both individually and in combination. To accomplish this, we trained a separate model on

every subset of the seven datasets, resulting in 127 unique models, each trained in eight replicates (Figure 3A; Table S3; see STAR Methods).

Given that smOOPs were longer than control transcripts, with a ~3-fold longer coding sequence (CDS) as the primary driver of the overall length enrichment (Figure S3B) and because all input tracks implicitly encode transcript length, we additionally trained a baseline model using sequence length as the sole feature, also in 8 replicates (Table S3). Transcript length alone achieves a good baseline performance (area under the receiver operating characteristic curve [AUROC] = 0.82, accuracy = 74%), which indicates that length provides meaningful information for smOOPs prediction (Figure 3B). The DL model trained only on sequence data greatly improved the prediction accuracy on the smOOPs previously not seen by the model (AUROC = 0.91, accuracy = 81%) (Figures 3B, S3C, and S3D), indicating that although length is a contributing factor, sequence-specific features provide critical information for accurately distinguishing smOOPs. Notably, the sequence-based model outperformed models trained on any other individual dataset (Figure 3B, number of datasets = 1). The second-highest predictive performance was achieved by models trained exclusively on global iCLIP data, with an AUROC of 0.89 and an accuracy of 79%, which may reflect the fact that both iCLIP and OOPS rely on UV crosslinking to detect RBP binding and therefore sample related biochemical events. Generally, as additional datasets were introduced, the predictive power gradually improved, ultimately reaching an AUROC of 0.94 and an accuracy of 83% with all datasets included (Figure 3B, number of datasets = 7). To confirm that the DL model accurately classifies smOOPs using unseen data from a different experimental batch, we repeated the semi-extractability assay and OOPS in nPSCs. The new batch produced a partially overlapping but not identical smOOPs set compared with the original dataset (Figure S3E; Tables S1 and S5), which served as an independent test set for model evaluation. We observed a reasonably strong generalization of the model to the new smOOPs and control datasets, with the all-features model achieving an AUROC of 0.79 (Figure S3F). Again, the predictive performance of the DL model trained on all features or on the sequence alone (AUROC of 0.75) exceeded the model trained on transcript length alone (AUROC of 0.67), emphasizing that the selected features are robust predictors across batches.

To highlight how each information layer contributed to the model's predictive power beyond the others, we assessed the average AUROC improvement by comparing the model's performance with and without each dataset. Our analysis revealed that the sequence, POSTAR3 peaks, global iCLIP, and PARIS-Intra layers significantly improved the performance of the models in which they were incorporated, giving us the confidence that these datasets contain important information about the features that distinguish smOOPs transcripts (Figure 3C). Surprisingly, RNAfold predictions reduced performance, presumably by introducing noise, due to the binary encoding of only the optimal structure, the limited ability of free energy minimization algorithms to predict secondary structures for long sequences,[49] and the absence of context-specific information (e.g., cellular

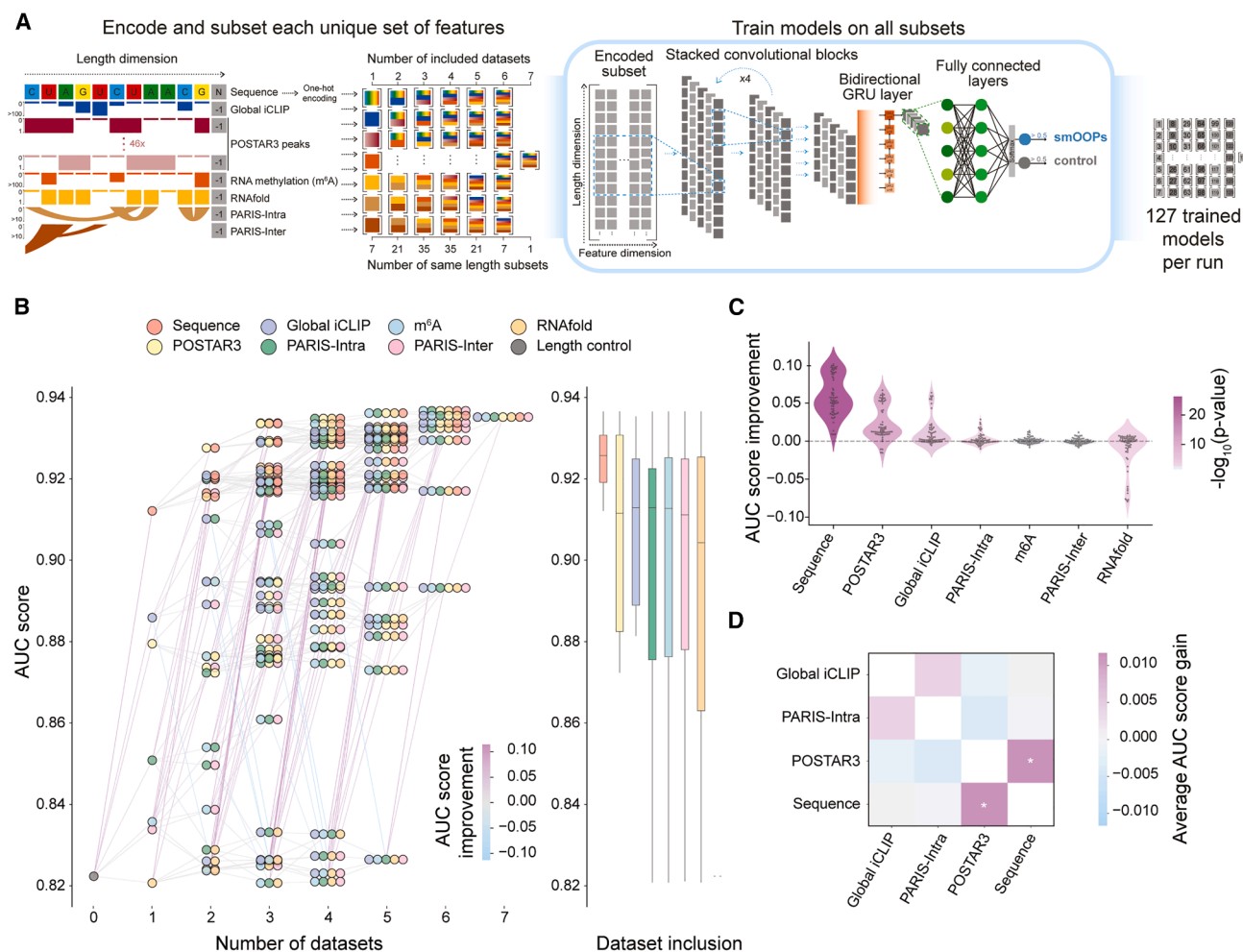

**Figure 3. Deep learning-based classification of smOOPs and control transcripts**

(A) Schematic visualization of the feature encoding and deep learning model training across 127 unique combinations of transcriptomic datasets. Each combination was used to train a model through convolutional and recurrent layers for nPSC smOOPs and control transcript classification.

(B) Model performance across dataset combinations. (Left) AUROC of the best model trained on each unique dataset combination. Dots represent included datasets, with lines connecting models where one is a subset of the other, color coded by AUROC improvement. (Right) Boxplots showing the median performance and interquartile range for all the models for which specific datasets were included.

(C) AUROC improvement with the addition of a particular dataset to each combination of previously included datasets. Color indicates the *p* value from a two-sided Welch's *t* test against zero.

(D) Pairwise feature combination analysis quantifying the difference between maximum individual and joined contributions of features to model performance. Statistically significant results are marked (*$p < 0.05$, two-sided Welch's *t* test against zero).

factors like RBPs or the influence of *in vivo* conditions that can affect folding). Most features showed minimal AUROC improvement when used in combination, likely because they provided limited additional information (Figure 3C). To explore this overlap further, we analyzed feature pairs from the top four most informative datasets (sequence, POSTAR3, global iCLIP, and PARIS-Intra) to assess their individual and combined contributions to AUROC. Excluding both features and adding one or both back revealed additive effects, particularly between RNA sequence and RBP occupancy. Notably, sequence data alone (which also reflect transcript length) encode most information represented by the other datasets, except for individual RBP binding profiles, which likely refined the model by distinguishing

sequences recognized by specific RBPs for better information extraction (Figure 3D). To validate the contribution of each feature across different transcript regions, we conducted masking experiments in which we omitted the 5′ untranslated region (5′ UTR), CDS, and 3′ untranslated region (3′ UTR) signal (Figure S3G). We found that the sequence data contributed most to the predictive power in the CDS, while the global iCLIP and POSTAR3 were most informative in the 3′ UTR. In contrast, the performance of the PARIS-Intra-based models decreased when either the CDS or the 3′ UTR was masked (Figure S3G).

Together, our analyses demonstrate that using the selected datasets within our DL framework enabled accurate smOOPs prediction and that unbiased training on all feature combinations

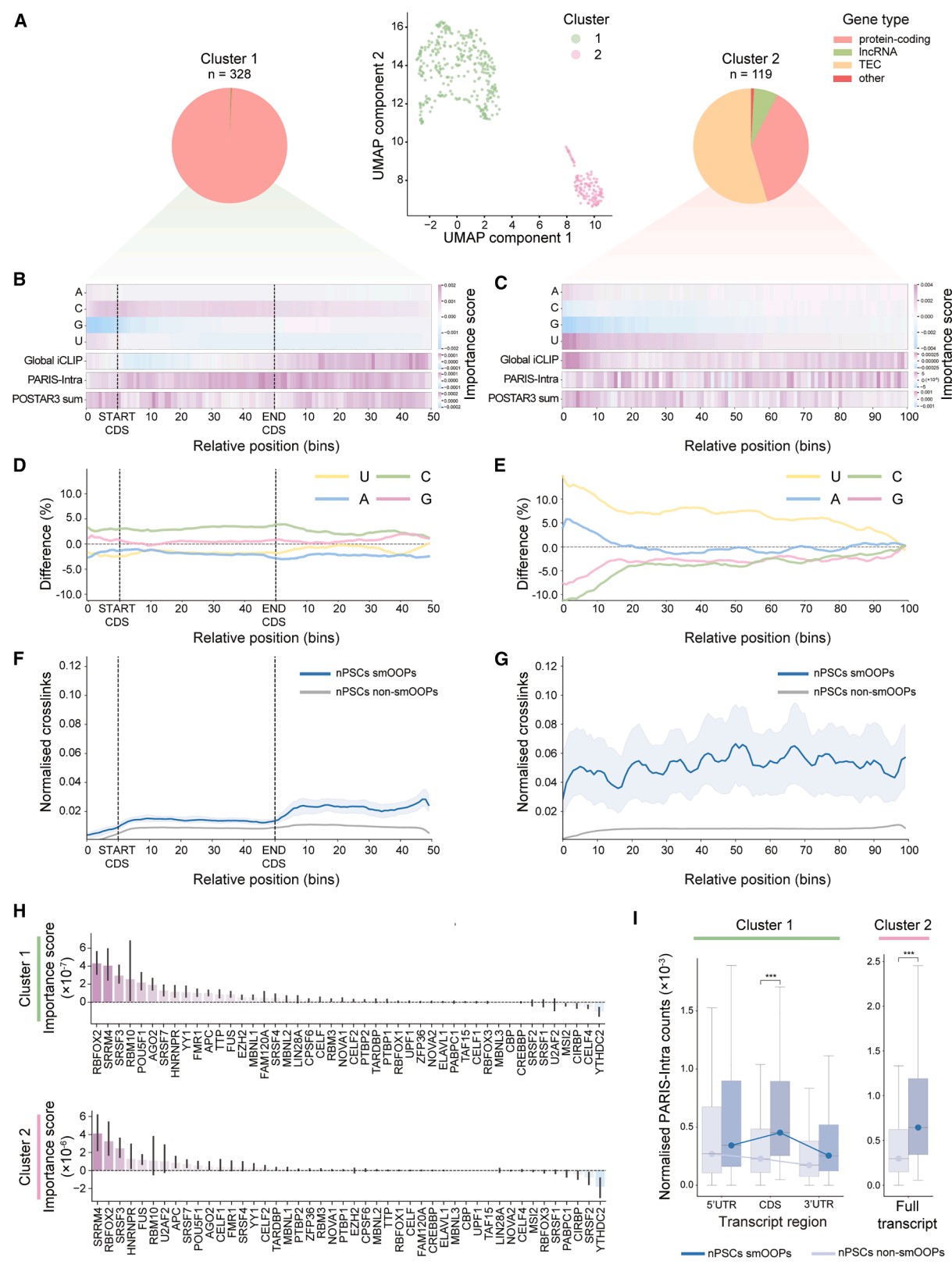

(legend on next page)

revealed nuanced dataset interactions predictive of these transcripts.

### Deconvolving the features of smOOPs

To extract the unique features distinguishing smOOPs, we used integrated gradients to compute nucleotide-resolution feature importance scores for the models trained on each of the top four datasets that improved the average model performance (i.e., sequence, POSTAR3, global iCLIP, and PARIS-Intra). Overlaying the importance scores over each transcript uncovered distinct patterns of feature contributions for each individual dataset (Figure S3H). We analyzed global feature patterns by dividing smOOPs transcripts into 100 bins and averaging feature importance scores within each bin, capturing patterns across datasets. We then clustered smOOPs based on the importance scores across these feature dimensions, revealing two clusters (cluster 1, $n = 328$; cluster 2, $n = 119$) (Figure 4A; Table S1).

Cluster 1 primarily contained mRNAs (98.2%), while cluster 2 was a mix of TECs (56.4%), mRNAs (36.4%), and lncRNAs (7.3%) (Figure 4A). We observed distinct sequence compositions in each cluster: cluster 1 exhibited a pronounced cytosine (C) enrichment across the transcript, particularly within the 5′ UTR and CDS, which overlapped with high PARIS-Intra importance (Figure 4B). In contrast, cluster 2 displayed increased overall importance of uridine (U), with adenine (A) being slightly more important at the 5′ ends of the transcripts (Figure 4C). Although global RBP occupancy (global iCLIP) is a key feature across all smOOPs, our analysis highlighted the positional specificity of RBP binding in each cluster: the importance of RBP occupancy was predominantly concentrated in the 3′ UTR for cluster 1, while for cluster 2, it was distributed more uniformly across the entire transcript (Figures 4B and 4C). Furthermore, the regions of POSTAR3 importance, summing individual CLIP binding profiles (POSTAR3 sum), portrayed more specific regions of importance, emphasizing the precise nature of RBP interactions at these sites (Figures S4A and S4B). Cumulatively, our DL approach pinpoints the key biological information for RNA condensation propensity, with sequence, intramolecular structure, and RBP binding information being the most predictive.

Comparing nPSC smOOPs with all non-smOOPs transcripts, we sought to validate and deepen the insights gained from our models by directly investigating features highlighted as important. Building on model predictions, a lower sequence complexity compared to controls was confirmed for both clusters: the mRNAs in cluster 1 portrayed a C-rich CDS (Figure 4D), with the "CCC" triplet being the most enriched, followed by "GCC," "CGC," "CCG," and "CCA" (Figures S4C and S4E). The most frequent triplets in the U-rich cluster 2 (Figure 4E) included "UUU," followed by "UUA," "UCU," "UGU," and "UAU" (Figures S4D and S4F). Global iCLIP data showed enhanced global RBP binding in the 3′ UTR of smOOPs in cluster 1 (Figure 4F); however, there was an overall 10-fold greater RBP occupancy for transcripts in cluster 2, and this was also more pronounced within the 3′ UTRs of cluster 2 mRNAs (Figures 4G and S4G). Since UV crosslinking preferentially targets uridines, we cannot determine the exact extent to which this bias influences the observed enrichment in RBP occupancy. Importance scores for POSTAR3 data revealed that in both clusters, RBFOX2, SRRM4, and SRSF3 carried key predictive information (Figure 4H). Since SRRM4 is not expressed at sufficient levels in our cell line, the model likely determined its importance based on the presence of its binding motifs. The frequency of intramolecular interactions determined by PARIS was also significantly increased in the CDS of smOOPs from cluster 1 and was overall higher in smOOPs from cluster 2 (Figure 4I). Despite these specific features of smOOPs, we did not observe any major differences in translation efficiency[50] or in mRNA stability[51] compared to non-smOOPs (Figures S4H–S4J).

These findings highlight that smOOPs are more strongly bound by RBPs, generally more structured than non-smOOPs, and can be divided into two clusters with distinct sequence composition in nPSCs: one comprising C-rich mRNAs and the other predominantly comprising A/U-rich transcripts. In both clusters, RBP binding plays a crucial role, emphasizing its role in shaping the behavior of these unique transcripts.

### smOOPs mRNAs encode proteins rich in IDRs

Since smOOPs in nPSCs were clearly distinguished from control transcripts based on their sequence features, we investigated

---

**Figure 4. Analysis of smOOPs predictive features**

(A) Uniform manifold approximation and projection (UMAP) of binned importance scores per feature for nPSC smOOPs. Each dot represents a transcript, color coded by cluster. The accompanying pie charts show the distribution of gene types within each cluster (TEC, to be experimentally confirmed).

(B) Heatmap showing the average nucleotide- and dataset-specific feature importance scores for all transcripts in cluster 1, divided into 10 bins for the 5′ UTR and 50 bins each for the CDS and 3′ UTR.

(C) Heatmap showing the average nucleotide- and dataset-specific feature importance scores for all transcripts in cluster 2, binned into 100 intervals along the transcript length.

(D) Per-bin difference in average nucleotide content between cluster 1 smOOPs and control transcripts, with nucleotide content divided into 10 bins for the 5′ UTR and 50 bins each for the CDS and 3′ UTR.

(E) Per-bin difference in average nucleotide content between cluster 2 smOOPs and control transcripts, with nucleotide content divided into 100 bins across the transcripts.

(F) Median global iCLIP signal, normalized for expression and binned (10 bins for the 5′ UTR and 50 bins each for the CDS and 3′ UTR; pooled from three replicates) for cluster 1 and control transcripts. Shaded areas represent 95% confidence intervals per bin, estimated via bootstrapping.

(G) Median global iCLIP signal, normalized for expression and binned (100 bins across the transcripts; pooled from three replicates) for cluster 2 smOOPs and control transcripts. Shaded areas represent 95% confidence intervals per bin, estimated via bootstrapping.

(H) Average importance scores for CLIP datasets from POSTAR3 across all transcripts in clusters 1 and 2, shown for each RNA-binding protein with 95% confidence intervals.

(I) Bar charts showing PARIS intramolecular hybrid counts across individual transcript regions for cluster 1 and smOOPs compared to all non-smOOPs transcripts, adjusted for region length and expression (***$p < 0.001$, two-sided Welch's $t$ test).

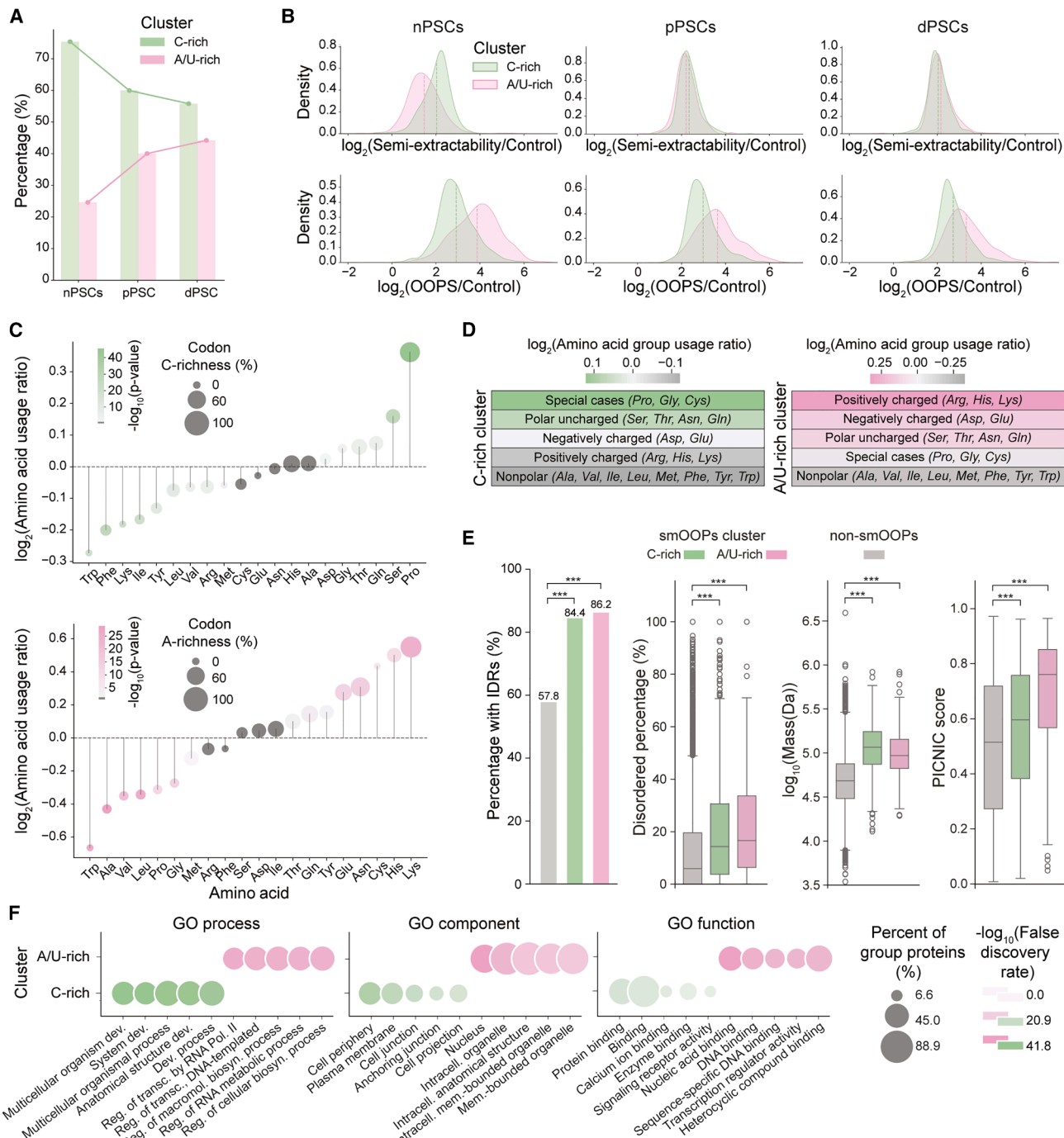

**Figure 5. IDR content, amino acid composition, and functional characteristics of smOOPs-encoded proteins**

(A) Percentage of smOOPs transcripts in the C-rich and A/U-rich clusters across developmental stages (nPSCs, pPSCs, and dPSCs).

(B) Density distributions of semi-extractability and OOPS enrichment for smOOPs transcripts in the C-rich and A/U-rich clusters across nPSCs, pPSCs, and dPSCs.

(C) Amino acid enrichment in C-rich and A/U-rich smOOPs clusters compared to non-smOOPs. Dot size indicates the percentage of C/A nucleotides in codons encoding each amino acid, with gradient strength showing statistical significance for the difference in mean codon usage (two-sided Welch's *t* test).

(D) Enrichment of amino acid groups in C-rich and A/U-rich smOOPs clusters compared to non-smOOPs.

*(legend continued on next page)*

whether these features persisted throughout development. To address this, we trained an additional DL model using only the sequence of smOOPs transcripts identified at all three developmental stages—nPSCs, pPSCs, and dPSCs—and compared them to the same set of common control transcripts (Table S1). This newly trained model achieved a performance comparable to that of the nPSC-only model (AUROC = 0.90, accuracy = 82%), indicating that sequence features uniquely define smOOPs throughout embryonic development, not only in nPSCs. Furthermore, when examining the model's learned features, we again identified two smOOPs clusters—C rich and A/U rich—which showed 96% consistency with those from the nPSCs-only model (Figures S5A and S5B; Table S1). This confirms that this separation is a general feature of smOOPs that persists throughout development.

Interestingly, the ratio of C-rich to A/U-rich smOOPs shifts during development. In nPSCs, C-rich smOOPs were predominant, comprising approximately 75% of nPSC-specific smOOPs. In dPSCs, their proportion declined to 56%, reflecting an increased representation of A/U-rich smOOPs at later developmental stages (Figures 5A and S5C). This observation prompted further investigation into how smOOPs behavior changes across semi-extractability and OOPS assays for the two clusters. While C-rich smOOPs maintained consistent enrichment across both assays, A/U-rich smOOPs became increasingly semi-extractable during development while showing reduced OOPS enrichment (Figure 5B). This coincided with a higher proportion of mRNAs in the A/U-rich cluster and a gradual decline of TECs (Figure S5D).

To elucidate the basis of the nucleotide composition differences between the smOOPs clusters, we examined the contribution of codon bias or amino acid composition. In line with the A/U-rich smOOPs nucleotide bias, the amino acid composition of this group was heavily enriched in charged and polar residues (Figures 5C and 5D). In contrast, C-rich smOOPs encoded proteins enriched in proline and polar uncharged residues, particularly serine (Figures 5C and 5D). Since many enriched amino acids (proline and serine in the C-rich cluster and glutamine, glutamate, and lysine in the A/U-rich cluster) are all highly abundant in the IDRs of proteins,[53] we investigated whether smOOPs encode proteins with an inherent propensity for condensation.

Structural analysis confirmed this hypothesis, revealing that both C-rich and A/U-rich clusters encode more proteins with IDRs and a higher percentage of disordered amino acids (Figure 5E). Notably, as many as 84.4% of C-rich and 86.2% of A/U-rich smOOPs-encoded proteins contain IDRs, while only 57.8% of non-smOOPs-encoded proteins contain IDRs (Figure 5E). In addition, smOOPs-encoded proteins are, on average, more than twice the size of the non-smOOPs group, resulting in significantly longer IDRs. Both the number of IDR-containing proteins and the proportion of disorder within these pro-

teins (Figure 5E) suggested that smOOPs-encoded proteins might be involved in condensate formation. We tested this using the PICNIC (proteins involved in condensates in cells) prediction model,[52] providing further evidence that smOOPs-encoded proteins are also more likely to be involved in condensates (Figures 5E and S5F). Observed protein features were reproducibly recapitulated among smOOPs in the second nPSCs batch (Figure S5E). Interestingly, despite their shared structural properties and condensation potential, the two clusters encode proteins with distinct cellular functions. Gene Ontology (GO) analysis revealed that C-rich cluster proteins are localized to the cell membrane and periphery, are mainly involved in developmental processes, and play roles in protein binding. In contrast, A/U-rich cluster proteins are mostly nuclear, involved in gene regulation and containing nucleic-acid-binding domains, especially zinc-finger motifs (Figures 5F and S5G; Table S4).

These findings suggest that smOOPs encode two classes of proteins with high condensation propensity but distinct cellular roles. Differences at both the RNA and protein levels underscore the sequence-driven and developmentally regulated nature of smOOPs, highlighting their involvement in condensate formation throughout development.

## DISCUSSION

Understanding the principles governing RNA condensates in their native cellular context is crucial to uncovering how they shape cellular biology. In this study, we use a dual methodology—semi-extractability assay and OOPS—to define smOOPs based on their shared biochemical properties during early murine embryonic development. This combination provides a comprehensive view of RNAs potentially involved in RNP assemblies and condensation. smOOPs are a distinct class of long transcripts that consist of predominantly protein-coding RNAs exhibiting subcellular localization in larger foci, among which many are enriched in known transcriptomes of RNP granules. RNA networks from RIC-seq further revealed that smOOPs form more densely connected subnetworks than expected, suggesting their proximity to one another and potential spatial organization and cellular compartmentalization.

We systematically investigate the characteristics of smOOPs using an integrative DL approach that identifies two clusters based on sequence composition, RBP-binding patterns, and structuredness: C-rich mRNAs with structured regions and extensive 3′ UTR RBP binding and A/U-rich transcripts with an overall higher RBP occupancy. A previous report has linked longer mRNAs and 3′ UTR-bound RBPs with local translation in subcytoplasmic compartments[37]; however, we detected no general differences in the translational output of smOOPs. This heterogeneity suggests that smOOPs may contribute to condensation through diverse mechanisms or in different cellular contexts. Across the three developmental time points,

(E) Percentage of proteins with IDRs and measures of disorder for C-rich and A/U-rich smOOPs clusters compared to non-smOOPs. Bar chart shows the percentage of proteins with IDRs, and the boxplots show the percentage of disorder in proteins, their mass, and PICNIC scores (proteins involved in condensates in cells)[52] (***$p < 0.001$; two-sided Wilcoxon rank-sum test for disorder percentage, two-sided Welch's $t$ test for mass and PICNIC scores).
(F) GO term enrichment analysis for C-rich and A/U-rich smOOPs clusters. The dot size represents the percentage of proteins enriched for each term, and the color intensity reflects statistical significance (false discovery rate [FDR]).

transcripts in the C-rich cluster show stable enrichment in both methods, while those in the A/U-rich cluster become increasingly semi-extractable and less OOPS enriched. This suggests that A/U-rich smOOPs undergo greater developmental changes in RNP assembly and may play distinct roles in condensate formation at later developmental stages.

The most striking finding is the link between smOOPs' RNA sequence features and the presence of IDRs in their encoded proteins. Disordered protein regions tend to be encoded by repetitive nucleotides or sequence motifs, such that the sequence repetitiveness is reinforced by codon biases.[54] Here, we show that both smOOPs clusters encode a significantly higher proportion of proteins with IDRs compared to non-smOOPs. However, the two clusters are distinguished by differences in nucleotide and amino acid composition, which likely contribute to the distinct functionalities of their encoded proteins. This finding hints at a possible coordination between RNA identity and the phase-separation potential of the proteins they encode, a concept that requires further systematic investigation.

While earlier studies have examined the roles of specific RNAs or RBPs in condensation, this study takes a broader approach by identifying and characterizing an entirely new class of RNAs with potential implications for phase separation. This resource and our findings provide a foundation for future research aimed at confirming the involvement of smOOPs in condensation, unraveling the functional relevance of the two identified clusters, and elucidating the mechanisms by which RNA features may coordinate with protein disorder and phase-separation potential. Such coordination has been described for nuclear speckles, where groups of proteins with condensation-prone domains promote the selective sequestration of related mRNAs encoding these proteins.[54] Comparison of smOOPs with high-throughput datasets of condensate-enriched RNAs revealed a significant over-representation of RNAs localized in stress granules. While long RNAs have been reported to localize in stress granules,[36,55,56] we also observed this enrichment in unstressed cells, raising the question of whether smOOPs act as early scaffolds for condensate formation.

Repeating the semi-extractability assay and OOPS years later in nPSCs confirmed that smOOPs are primarily defined by shared features, which remained consistently predictive despite batch-to-batch differences in gene-level identities. Our definition of smOOPs used stringent thresholds to capture strong effect sizes and ensure that the identified RNAs exhibited robust semi-extractability and OOPS enrichment. Nonetheless, these properties lie on a continuum rather than forming discrete categories, so fixed cutoffs inevitably exclude RNAs with intermediate characteristics. Thus, we view smOOPs as an assay-defined subset with consistent features, though their exact membership may vary across experiments due to biological or technical variability.

In terms of methodology, our study demonstrates the power of explainable machine learning to reveal complex patterns across diverse datasets, enabling unbiased classification and characterization of gene groups. This is especially valuable in the study of condensates, where distinguishing different assemblies and RNA functions is challenging using traditional approaches. By integrating RNA proximity-ligation datasets into network-based analyses, we provide an approach to uncover new insights into the features organizing specific RNA networks, aiding our understanding of protein-RNA condensates.

Overall, this study opens a new avenue for understanding the complex interplay between RNA identity and protein condensation potential in cellular organization and function, positioning smOOPs as potential players in the regulation of condensation processes.

### Limitations of the study

This study presents a new class of semi-extractable RNAs with high RBP occupancy determined by both OOPS and global iCLIP. Both assays rely on UV crosslinking, which has a strong bias toward uridines. Therefore, we cannot determine to what extent the observed higher RBP occupancy, especially in the A/U-rich cluster, reflects the actual increase in biological interactions or increased crosslinking efficiency. Although we performed orthogonal HCR-FISH experiments that have been used as proxies for condensation, these did not directly assess phase-separation processes.

smOOPs networks obtained by RIC-seq rely on pairwise RNA proximity, showing that individual smOOPs can be near each other, though not necessarily all at the same time. Furthermore, it remains unclear whether their non-random association and proximity result from RNA condensation processes (such as co-assembly) or if they are influenced by other factors. GO term analysis suggests different functions of smOOPs-encoded proteins from C- and A/U-rich clusters; however, the functional implications in the context of development and condensate formation could be further explored.

### RESOURCE AVAILABILITY

#### Lead contact

Further information and requests for resources and reagents should be directed to and will be fulfilled by the lead contact, Miha Modic (miha.modic@kit.edu).

#### Materials availability

The strains generated in the course of this work are freely available to academic researchers through the lead contact.

#### Data and code availability

Newly produced and publicly available data were used for this work. Newly produced data were deposited on ArrayExpress under accession numbers E-MTAB-14762 (RNA-seq for semi-extractability and OOPS assays in nPSCs, pPSCs, and dPSCs), E-MTAB-15428 (RNA-seq for semi-extractability and OOPS assays in nPSCs, second batch), E-MTAB-14763 (global iCLIP), and E-MTAB-14764 (RIC-seq). Raw PARIS data are available at GEO: GSE74353. Raw ribosome profiling data are available at GEO: GSE30839, with processed data (translation efficiency values) obtained from the supplemental information.[50] Ribosome profiling data on mESCs were downloaded from GEO (accession numbers GEO: GSM3943973 and GSM3943975). SlamSeq metabolic RNA-seq data were downloaded from GEO: GSE99978. Raw miCLIP (m6A) data are available at GEO: GSE169549. POSTAR3 CLIP datasets were obtained from the POSTAR3 platform (http://postar.ncrnalab.org/). The best-performing models trained for this manuscript are deposited at Zenodo (https://doi.org/10.5281/zenodo.17076365). The PICNIC scores for the mouse proteome were obtained from the PICNIC platform (https://picnic.cd-code.org/). The lncRNA prediction scores were calculated using lncRNAnet (https://github.com/nofundamental/lncRNAnet). The code and notebooks to analyze the data and produce the figures in this

work are available at https://github.com/ModicLab/smOOPs_project. The images and code used for HCR-FISH quantification are deposited at Zenodo (https://doi.org/10.5281/zenodo.13860869).

## ACKNOWLEDGMENTS

We would like to thank Flora Lee for establishing the RIC-seq library preparation method and Oscar Wilkins for designing guides for rRNA depletion using Ribocutter and helping with the Ultraplex demultiplexing tool. We are grateful to Charlotte Capitanchik, Moritz Kreysing, Lennart Hilbert, Tatjana Trček, and the lab members for useful comments on the manuscript. We would like to thank Eneko Villanueva for providing us with a detailed protocol for OOPS, Yuanchao Xue for a detailed RIC-seq protocol, and Joel Ryan for continuous advice on HCR-FISH imaging, together with Petra Čotar and Luka Čehovin Zajc for automated image acquisition. We thank Thomas H. Kapral and the other POSTAR3 authors for access to the full CLIPdb dataset. We also thank Tine Tesovnik and Jernej Kovač for their assistance and sequencing at the University Children's Hospital at Ljubljana University Medical Center. We also wish to acknowledge the Advanced Sequencing Facility and Nemo HPC at the Francis Crick Institute and HPC VEGA at the IZUM, the Institute of Information Science. This work was supported by Wellcome (218672/Z/19/Z, 225081/Z/22/Z, and 215593); the Slovenian Research Agency (J4-50145, J4-60070, J7-2596, N1-0240, and P1-0060); the Carl Zeiss Foundation and the Center for Synthetic Genomics (CZS Center SynGen); the Helmholtz Program "Natural, Artificial, and Cognitive Information Processing"; a Janko Jamnik PhD Fellowship and L'Oréal-UNESCO For Women in Science Award awarded to T.K.; the European Research Council (ERC) under the EU's Horizon 2020 research and innovation program (grant agreement no 835300); the Johanna Quandt Young Academy fellowship; the Francis Crick Institute, which receives its core funding from Cancer Research UK (CC0102), the UK Medical Research Council (CC0102), and Wellcome (CC0102); and the UK Dementia Research Institute (award number UK DRI-RE21605) through UK DRI, Ltd., principally funded by the UK Medical Research Council. HPC VEGA is financed through the HPC RIVR consortium (www.hpc-rivr.si) and EuroHPC JU (eurohpc-ju.europa.eu). The funders had no role in the study design, data collection and analysis, decision to publish, or preparation of the manuscript.

## AUTHOR CONTRIBUTIONS

Conceptualization, T.K., J.N., I.A.I., and M.M.; investigation, T.K.; project administration, T.K., I.A.I., and M.M.; formal analysis, T.K., J.N., I.A.I., B.K., and D.M.J.; writing – original draft, T.K., J.N., I.A.I., and M.M.; writing – review & editing, T.K., J.N., I.A.I., A.M.C., J.U., and M.M.; methodology, J.N. and I.A.I.; visualization, T.K., J.N., and I.A.I.; data curation, J.N. and I.A.I.; software, I.A.I., B.K., I.U., D.M.J., and A.M.C.; supervision, I.A.I., A.M.C., N.M.L., J.U., and M.M.; resources, N.M.L., J.U., and M.M.; funding acquisition, N.M.L., J.U., and M.M.

## DECLARATION OF INTERESTS

The authors declare no competing interests.

## STAR★METHODS

Detailed methods are provided in the online version of this paper and include the following:

- KEY RESOURCES TABLE
- EXPERIMENTAL MODEL AND STUDY PARTICIPANT DETAILS
  - Cell culture
- METHOD DETAILS
  - Semi-extractability and OOPS assays
  - Global iCLIP
  - RIC-seq
  - HCR-FISH
- QUANTIFICATION AND STATISTICAL ANALYSIS
  - HCR-FISH image analysis
  - Reference annotation
  - RNA-seq data analysis
  - Global iCLIP data analysis
  - RIC-seq data processing
  - RIC-seq network inference and analysis
  - PARIS data processing
  - Ribo-seq data processing
  - Processing of RNP granule gene sets
  - lncRNAnet prediction of long non-coding RNA
  - Data preparation for deep learning
  - Data encoding
  - Deep learning model architecture
  - Training of the powerset
  - Assessing model performance
  - Explaining the trained weights
  - Gaining insight into the learned features
  - Validation of the predicted importance
  - Investigation of the protein features

## SUPPLEMENTAL INFORMATION

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

## Article

**CellPress**

## STAR★METHODS

### KEY RESOURCES TABLE

| REAGENT or RESOURCE | SOURCE | IDENTIFIER |
|---|---|---|
| **Chemicals, peptides, and recombinant proteins** | | |
| TRIzol | Invitrogen | Cat#15596018 |
| TRIzol-LS | Invitrogen | Cat#10296010 |
| Proteinase K | Roche | Cat#3115828001 |
| Phenol:Chloroform:Isoamyl Alcohol | Sigma | Cat#P3803 |
| Agencourt AMPure XP | Beckman Coulter | Cat#A63881 |
| RNase I | Thermo Scientific | Cat#EN0602 |
| NuPAGE® Novex 4–12% Bis-Tris Gel 1.0 mm, 12 well | Thermo Fisher | Cat#NP0322BOX |
| 6% TBE-Urea gel | Thermo Fisher | Cat#EC6865BOX |
| Disuccinimidyl suberate (DSS) | Thermo Fisher | Cat#21655 |
| Micrococcal Nuclease (MNase) | ThermoFisher | Cat#EN0181 |
| T4 RNA ligase | NEB | Cat#M0204L |
| FastAP Thermosensitive Alkaline Phosphatase | Thermo Scientific | Cat#EF0651 |
| T4 Polynucleotide Kinase (PNK) | NEB | Cat#M0201L |
| RecJf Exonuclease | NEB | Cat#M0264S |
| T4 DNA Ligase | Thermo Fisher | Cat#EL001 |
| MyOne Streptavidin C1 beads | Invitrogen | Cat#65001 |
| Nuclease-free water | Thermo Fisher | Cat#AM9930 |
| Turbo DNase | Thermo Fisher | Cat#AM2238 |
| Phusion™ High-Fidelity PCR Master Mix with HF Buffer | NEB | Cat#M0531L |
| Geltrex™ Reduced-Growth Factor Basement-Membrane Matrix, LDEV-free, stem-cell qualified | Gibco | Cat#A1413302 |
| μ-Slide 8 Well Glass Bottom | Ibidi | Cat#80827 |
| Fluoromount G | Thermo Scientific | Cat#00-4958-02 |
| Pierce™ 16% Formaldehyde (w/v), Methanol-free | ThermoFisher | Cat#28906 |
| **Critical commercial assays** | | |
| RNeasy Plus Mini Kit (Qiagen) | Qiagen | Cat#74134 |
| CORALL Total RNA-Seq V1 with RiboCop rRNA depletion | Lexogen | Cat#95.96 |
| pCp-biotin ligation kit | Thermo Fisher | Cat#20160 |
| Superscript IV RT kit | Invitrogen | Cat#18090050 |
| **Deposited data** | | |
| Total RNA-seq for semi-extractability and OOPS assays | This paper | ArrayExpress (E-MTAB-14762; E-MTAB-15428) |
| Global iCLIP | This paper | ArrayExpress (E-MTAB-14763) |
| RIC-seq | This paper | ArrayExpress (E-MTAB-14764) |
| Raw images and image analysis scripts | This paper | https://doi.org/10.5281/zenodo.13860869 |
| Trained models (with highest performance) | This paper | https://doi.org/10.5281/zenodo.17076366 |
| Ribo-seq dataset | Ingolia et al., 2011[50] | GEO, Accession number: GSE30839 |
| Ribosome profiling data (monosome and disome) | Tuck et al., 2020[41] | GEO, Accession numbers: GSM3943973 and GSM3943975 |
| SLAM-seq dataset | Herzog et al., 2017[51] | GEO, Accession number: GSE99978 |
| miCLIP (m6a) dataset | Modic et al., 2024[45] | GEO, Accession number: GSE169549 |
| POSTAR3 collection | Zhao et al., 2022[48] | http://postar.ncrnalab.org |

*(Continued on next page)*

**Cell Genomics**
**Article**

*Continued*

| REAGENT or RESOURCE | SOURCE | IDENTIFIER |
|---|---|---|
| UniProt | The UniProt Consortium[57] | https://www.uniprot.org |
| STRING version 12.0 | Szklarczyk et al., 2023[58] | https://string-db.org |
| Original code for data analysis | This paper | https://github.com/ModicLab/smOOPs_project |
| **Experimental models: Cell lines** | | |
| *M. musculus:* WT pluripotent stem cells IDG3.2 (129S8/B6 background) | Hitz et al., 2007[59] | (RRID:CVCL_A2WN) |
| *M. musculus:* TGFP/+;Foxa2tagRFP/+ (G9) on IDG3.2 background | Scheibner et al., 2021[60] | |
| **Oligonucleotides** | | |
| Oligonucleotides | This paper | Table S4 |
| **Software and algorithms** | | |
| Fiji | Schindelin et al., 2012[61] | https://fiji.sc/ |
| Rstudio v2024.12.1 | – | http://www.posit.co/ |
| R version 4.3.2 | R Core Team | https://www.r-project.org/ |
| Python 3.9 | – | https://www.python.org |
| iCount-Mini | – | https://github.com/ulelab/icount-mini |
| Tosca v1.0.0 | Chakrabarti et al., 2023[62] | https://github.com/amchakra/tosca |
| nf-core/rnaseq v3.4 | Bushnell et al., 2017[63]; Ewels et al., 2020[64] | https://nf-co.re/rnaseq/3.4 |
| R version 4.4.0 | R Core Team | https://www.r-project.org/ |
| DESeq2 v1.44.0 | Love et al., 2014[65] | https://bioconductor.org/packages/release/bioc/html/DESeq2.html |
| Ashr | Stephens, 2017[66] | https://cran.r-project.org/package=ashr |
| ComplexHeatmap v2.20.0 | Gu et al., 2016[67] | https://bioconductor.org/packages/release/bioc/html/ComplexHeatmap.html |
| Ultraplex v1.2.9 | Wilkins et al., 2021[68] | https://github.com/ulelab/ultraplex |
| nf-core/clipseq v1.0.0 | West et al., 2023[69] | https://nf-co.re/clipseq/1.0.0 |
| Cutadapt | Martin, 2011[70] | https://cutadapt.readthedocs.io/en/stable/ |
| BBMerge | Bushnell et al., 2017[63] | https://jgi.doe.gov/data-and-tools/bbtools/bb-tools-user-guide/bbmerge/ |
| featureCounts | Liao et al., 2013[71] | http://bioinf.wehi.edu.au/featureCounts/ |
| igraph v2.0.3 | Csardi & Nepusz, 2006[72,73] | https://igraph.org/ |
| bedtools | Quinlan & Hall, 2010[74] | https://bedtools.readthedocs.io/en/latest/ |
| RNAfold | Lorenz et al., 2011[47] | http://rna.tbi.univie.ac.at/cgi-bin/RNAWebSuite/RNAfold.cgi |
| lncRNAnet | Baek et al., 2018[40] | https://github.com/nofundamental/lncRNAnet |
| TensorFlow 2.12.0 | Abadi et al., 2015[75] | https://www.tensorflow.org |
| UMAP | McInnes and Healy, 2018[76] | https://umap-learn.readthedocs.io |
| Optuna framework | Akiba et al., 2019[76] | https://github.com/optuna/optuna |
| riboseq-flow v1.1.1 | Iosub et al., 2024[77] | https://github.com/iraiosub/riboseq-flow |
| RiboCode 1.2.14 | Xiao et al., 2018[78] | https://github.com/xryanglab/RiboCode |
| clipplotr | Chakrabarti et al., 2023[79] | https://github.com/ulelab/clipplotr |
| ComplexUpset 1.3.3 | – | https://cran.r-project.org/web/packages/ComplexUpset/index.html |
| circlize 0.4.16 | Gu et al., 2014[80] | https://cran.r-project.org/package=circlize |
| corrplot 0.94 | – | https://cran.r-project.org/package=corrplot |
| **Other** | | |
| UV Crosslinker CL-3000 (254 nm) | AnalytikJena | Cat# 849-95-0615-02 |

## EXPERIMENTAL MODEL AND STUDY PARTICIPANT DETAILS

### Cell culture

Low-passage, wildtype mouse pluripotent stem cells IDG3.2 PSCs (129S8/B6 background)[59] and TGFP/+;Foxa2tagRFP/+ (G9)[60] were cultured feeder-free, on cell culture dishes (TPP) coated with 0.1% Gelatin (ES-006-B, Millipore). Naive PSCs (nPSCs) were maintained in N2B27 medium composed of 1:1 Neurobasal (21103049) and DMEM-F12 (11320074) medium containing N2 (17502001) and B27 (17504001) supplements, 1% Glutamax (35050061), 1% nonessential amino acids (11140050) and 0.1 mM 2-mercaptoethanol (31350-010) (all Thermo Fisher Scientific), 12 ng/mL LIF (Qk018, Qkine), with additional use of small molecule inhibitors: for a condition commonly named 2iLIF, 1 μM MEK inhibitor PD0325901 (1408, Axon Medchem) and 3 μM GSK3 inhibitor CHIR99021 (SML1046, Sigma). nPSCs were fed every day and split every 2–3 days using Accutase (A6964, Sigma). To transit from naive to primed pluripotency state, the G9 PSCs were accutased and seeded onto gelatin and FBS-coated (EmbryoMax ES Cell Qualified FBS, ES-009-B, Merck) plates in N2B27 medium supplemented with 1,000 U/mL LIF (ESGRO ESG1107, Merck), 12 ng/mL bFGF (100-18B, Peprotech), 20 ng/mL Activin A (338-AC-050, R&D Systems), and 2 μM IWP2 (3533, Tocris). Once the nPSCs reached primed pluripotency state - pPSCs - they can be maintained indefinitely using the abovementioned FAI medium. For further expansion, pPSCs G9 cells were maintained in FAI medium on cell culture dishes (TPP) coated with Geltrex (A1413302, Gibco), fed every day and split every 2–3 days using Accutase (A6964, Sigma).

To generate differentiated primitive streak progenitors (dPSCs) pPSCs were grown in Wnt3a-differentiation medium (N2B27 medium supplemented with 20 ng/mL ActA, 12 ng/mL bFGF and 250 ng/mL Wnt3a) for 24 h prior collecting the cells.

## METHOD DETAILS

### Semi-extractability and OOPS assays

pPSCs G9 (p9) were prepared for experiment one day prior harvesting by plating them on 10 cm plates (TPP), and Wnt3a-differentiation medium was added to pPSCs for 24 h to generate differentiated primitive streak progenitors (dPSCs). For this experiment only, 5% FBS (ES-009-B, Merck) was added to gelatin during coating for nPSCs (p20). Cells were prepared as follows: for semi-extractability and standard TRIzol RNA-seq controls one confluent 6-well was used per replicate, triplicates in total. After washing cells 2× with PBS, 1 mL TRIzol-LS (10296010, Invitrogen) was added to the plate and scraped, then transferred to a 1.5 mL Eppendorf tube. Samples were snap-frozen and stored at −80°C. The semi-extractability assay was performed according to the protocol[20]: upon thawing and diluting TRIzol LS accordingly, the sample was heated for 10 min at 55°C and then sheared 40× through a 20G needle (the heating and shearing steps were omitted for control samples). Next, chloroform was added and mixed vigorously before the tube was centrifuged at 12,000× g for 15 min at 4°C. The aqueous phase was then mixed with 1.5× volume of 100% ethanol, and the isolation was continued with the RNeasy Plus Mini Kit (Qiagen) according to manufacturer's instructions.

For OOPS, one confluent 10 cm plate was used per replicate (triplicates in total), washed twice with ice-cold PBS before cross-linking them at 400 mJ/cm² (Crosslinker CL-3000 at 254 nm (AnalytikJena)). After removing residual PBS, 1 mL TRIzol-LS was added to the plate before scraping off the cells content and transferring it to a 1.5 mL Eppendorf tube that was stored at −80°C. For RNA isolation, we followed the protocol provided by the Lilley Lab.[81] Briefly, the content of TRIzol LS tubes was first diluted with 1× volume nuclease-free water, then we added 1/5th volume of chloroform, mixed vigorously and centrifuged at 12,000× g for 15 min at 4°C. The aqueous and organic phase were carefully removed and the remaining interphase resuspended in fresh 1 mL TRIzol (15596018, Invitrogen). Again, 200 μL chloroform was added before centrifugation and removal of aqueous and organic phases. Finally, the interphase was resuspended in 1 mL TRIzol once more before adding chloroform and separating the phases by centrifugation (12,000× g, 15 min, 4°C). This time, we removed as much of the aqueous and organic phases as possible and precipitated the remaining interphase (100 μL) using 100% Methanol (1 mL). After vortexing, the sample was centrifuged (14,000× g for 10 min at room temperature), supernatant was removed and the pellet digested using proteinase K (sro-3115828001, Roche) for two hours at 50°C. RNA was finally extracted by addition of 1 volume of phenol:chloroform (P3803, Sigma) and vortexing before centrifugation at 10,000× g for 10 min at room temperature. The aqueous phase was collected and RNA was precipitated for 20 min at room temperature with isopropanol. After centrifugation (12,000×g, 10 min, 4°C) the RNA pellet was washed twice using 75% ethanol and finally resuspended in nuclease-free water (AM9930, Thermo Fisher). All RNA samples were treated with Turbo DNase (AM2238, Thermo Fisher) for 30 min at 37°C to remove DNA contaminants and bead-purified using Agencourt AMPure XP (A63881, Beckman Coulter). Sequencing libraries were prepared using CORALL Total RNA-Seq V1 with RiboCop rRNA depletion (Lexogen) starting with 500 ng DNase-treated RNA as input. The resulting cDNA libraries were sequenced as single-end 100 bp reads on HiSeq at the Advanced Sequencing Facility at The Francis Crick Institute.

For the second batch of semi-extractability and OOPS, nPSCs G9 (p30) were cultured on gelatin alone (ES-006-B, Millipore) and prepared the same way as described above with a few adjustments. TRIzol (15596018, Invitrogen) instead of TRIzol-LS was used for RNA extraction and the library was prepared using CORALL Total RNA-Seq Kit with RiboCop (V2) starting with 1000 ng DNase-treated RNA as input. The resulting cDNA libraries were sequenced as paired-end 150 bp reads on Illumina NovaSeq X Plus Series with only Read 1 used for analysis (Table S5).

### Global iCLIP

iCLIP was performed according to the iiCLIP protocol.[42] First, 10 cm plates of nPSCs (G9, passage 17) and pPSCs (G9, passage 20) containing ~1.5 mg of total protein mass per sample were lysed. Three replicates were prepared for each developmental stage. Briefly, cells were lysed upon UV-crosslinking (150 mJ/cm² in Crosslinker CL-3000 at 254 nm (AnalytikJena)) and treated with RNase I before binding the RNA to an IR dye-labelled adaptor and separating the RNA-protein complexes on SDS-PAGE. After cutting the desired part of the membrane (from 40 kDa upwards), we digested the proteins and extracted the protein-bound RNA to prepare libraries for next-generation-sequencing (see Table S6 for adapter and primer sequences). Libraries were sequenced as paired-end 150 bp reads on NovaSeq at Clinical Institute of Special Laboratory Diagnostics, University Children's Hospital at Ljubljana University Medical Center.

### RIC-seq

Experiments were performed using nPSCs (G9, passage 20) and pPSCs (G9, passage 13)) in three replicates following the protocol[82] with some modifications. Two 10 cm confluent plates per sample were crosslinked using freshly prepared 0.5 mg/mL DSS (21655, Thermo Fisher Scientific) at room temperature for 30 min (rotating at 20 rpm). The reaction was quenched using 20 mM Tris (pH7.5) for 15 min at room temperature (rotating at 20 rpm). All centrifugation steps were carried out at 3500 rpm for 5 min at 4°C. After pelleting the cells, they were washed with PBS and then permeabilized (10 mM Tris-HCl (pH 7.5), 10 mM NaCl, 0.5% Igepal, 0.3% Triton X-100 and 0.1% Tween 20) for 15 min on ice. Cells were washed three times with 1× PNK buffer before fragmenting the RNA using 6U MNase (EN0181, ThermoFisher) at 37°C for 10 min. The following steps including FastAP treatment (EF0651, Thermo Fisher), pCp-biotin ligation (20160, Thermo Fisher), second FastAP and PNK treatment (M0201L, NEB) were performed as in the original protocol. Proximity ligation was performed using a different T4 RNA ligase (M0204L, NEB) and hence 1 mM ATP was added to the overnight reaction. Finally, cells were lysed using 200 μL proteinase K buffer (10 mM Tris-HCl pH 7.4, 100 mM NaCl, 1mM EDTA, 0.2% SDS) and 50 μL proteinase K (sro-3115828001, Roche) and an addition of 1.5 μL Turbo DNase (AM2238, Thermo Fisher), incubating at 37°C for 30 min, then 50°C for 60 min before adding Trizol-LS and snap freezing the samples. Once thawed and brought to room temperature, the samples were heated for 10 min at 55°C before adding the chloroform and precipitating RNA from the aqueous phase with isopropanol. The RNA was treated with Turbo DNASe once more and cleaned up using phenol:chloroform extraction as above. 21 μg of RNA was fragmented using 5× First Strand Synthesis Buffer (SuperScript IV, 18090050, Invitrogen) for 3.5 min at 94°C, immediately placed on ice and mixed with the MyOne Streptavidin C1 beads (65001, Invitrogen) to pull down biotinylated RNA (30 min at room temperature). Eluted RNA (10 μL) was extracted using phenol:chloroform method (see above). The biotinylation and pulldown efficiency were confirmed using dot blot assay before preparing sequencing libraries.

The biotin-enriched eluate was next subjected to 3′end dephosphorylation using PNK (M0201L, NEB) and FastAP (EF0654, Thermo Fisher) and purified using Agencourt AMPure XP beads (A63881, Beckman Coulter) before 3′end adapter ligation overnight at 20°C (Table S7). Once again, RNA cleanup was done with Agencourt AMPure XP beads before adapter removal (using Deadenylase (M0331, NEB) and RecJf exonuclease (M0264S, NEB)). Reverse transcription was performed according to the Superscript IV RT kit (18090050, Invitrogen) manual with the use of a custom RT primer (Table S7). Following cDNA cleanup (Agencourt AMPure XP beads) 5′ cDNA adapter was ligated using T4 DNA ligase (EL001, Thermo Fisher, without ATP) (Table S7). Samples were loaded onto 6% TBE-Urea gel (EC6865BOX, Thermo Fisher) and cDNAs exceeding 200 nt were excised from the gel and extracted using Crush-Soak Gel buffer (as per iiCLIP protocol[42]), followed by phenol:chloroform extraction. Precipitated cDNAs were stored at −20°C before performing PCR using P5/P7 standard Illumina primers and Phusion HF master mix (M0531L, NEB). Ribosomal RNA contaminants were removed from the final library using Ribocutter which utilises Cas9-guided rRNA depletion[83]; for this, 275 gRNAs were designed against mature 5S, 18S and 28S rRNAs (obtained as 50 pmol oPool from IDT) (Table S8). Final libraries (~10 nM) were treated with 4 μM sgRNAs for 30 min at 37°C and after beads purification, the libraries were reamplified with additional 6 cycles. They were sequenced as paired-end 150 bp reads on NovaSeq at Clinical Institute of Special Laboratory Diagnostics, University Children's Hospital at Ljubljana University Medical Center.

### HCR-FISH

Hybridization-chain reaction FISH was prepared by following the protocol[43] with slight modifications. The probes (8 pairs of probes per target, Table S9) were designed using https://github.com/rwnull/insitu_probe_generator[84] against mature mRNAs, however not targeting exon-exon junctions (average expression for controls 23.6 ± 16 TPM, for smOOPs 37.9 TPM ±31 TPM in semi-extractability total RNA-seq library). For imaging purposes WT IDG3.2 nPSCs were plated on Geltrex (A1413302, Gibco) coated 8-well glass-bottom ibidi plate (80827, ibidi) one day prior to fixation. After washing the cells with PBS, the cells were fixed by the fixation mixture (4% formaldehyde, 0.4% glyoxal, 0.1% methanol, 1× PBS). Amplification stage of a protocol to generate a tethered fluorescent amplification polymer lasted 10 h. Cells were then washed as described in the protocol and finally mounted in 300 μL Fluoromount G (00-4958-02, Thermo Scientific). For each transcript, over 70 nuclei were imaged and over 890 RNA foci counted. Images were acquired with a custom-built STED microscope (Abberior instruments) using a 1.2 NA 60× water immersion objective and lasers running at 80 MHz repetition rate. We excited fluorescently labeled mRNA by one of the three lasers at either 488, 561 or 640 nm, with 120 ps pulse length, with maximal power of 116 μW, 111 μW and 300 μW in the sample plane and DAPI stained nuclei with 405 nm laser with maximal power 810 mW in the sample plane. The laser powers used were 10% for 405 laser and 30% for the other three lasers. We acquired the fluorescence intensity using an avalanche photodiode with 500–550 nm, 580–625 nm or 650–720 nm

filters (Semrock) in front. The combinations of lasers and detectors were as follows: 405 nm laser and 500–550 nm, 488 nm laser and 500–550 nm, 561 nm laser and 680-620 nm filter, and 640 nm laser and 650–720 nm filter. The dwell time in the pixel was 10 μs, the pixel size was set to 50 nm and the pinhole size was set to 1.07 AU to achieve a good confocal resolution. Images were acquired with the help of automatic acquisition as described by Trupej and colleagues.[85]

## QUANTIFICATION AND STATISTICAL ANALYSIS

### HCR-FISH image analysis

The image analysis was done in Fiji[61] using custom-made macros. Briefly, for each fluorophore background was subtracted (30.0 pixels rolling radius) and threshold was set accordingly to conform all different transcripts with the same fluorophore ((9,255) for channel for 561, (3,255) for the 488 channel and (33,255) for the 640 channel). Next, particle analysis was performed and all subsequent analysis was done using "Fiji's particle analyser" and R (version 4.3.2, http://www.posit.co/). Total intensity was calculated by multiplying mean intensity and area for each foci. The average of total intensity for the control transcripts in each fluorophore was used to normalise other values and obtain "Total intensity".

### Reference annotation

For all analyses we used the GRCm39 build of the mouse genome with the Gencode M27 annotation. We used a custom reference sequence built on this annotation for the alignment of hybrid reads, generated as previously described.[62] To unambiguously annotate the genes within hybrid reads, we used a flattened annotation produced by iCount-Mini (https://github.com/ulelab/icount-mini). Both are available for download with the Tosca pipeline[62] (https://github.com/amchakra/tosca).

### RNA-seq data analysis

The sequencing reads were processed using nf-core/rnaseq version 3.4[63,64] (https://nf-co.re/rnaseq/3.4). For differential expression analyses we used gene-summarised count tables generated by nf-core/rnaseq 3.4 (using Salmon) as input to DESeq2 version 1.44.0.[65] The design incorporated both stage (nPSCs, pPSCs, dPSCs) and assay (control, semi-extractability, OOPS) factors. Prior to running DESeq2, we pre-filtered the count matrix to retain only genes with at least 10 normalised counts in at least 6 samples. For each desired contrast, we extracted results using the Wald test and applied ashr Log2FoldChange (LFC) shrinkage.[66] To identify genes enriched in OOPS or semi-extractability compared to control, we selected genes with padj <0.01 and LFC >1 for each contrast (semi-extractability vs. control or OOPS vs. control at each stage). Genes passing these thresholds in both semi-extractability and OOPS were defined as smOOPs. For data visualisation across assays, we converted rlog-normalised count data into Z-scores for each stage and plotted it using the ComplexHeatmap package[67] version 2.20.0. For the second batch described in Figure S3, smOOPs were defined as genes with padj <0.01 and LFC >0 in both assays. We used a reduced LFC threshold to account for the lower magnitude of fold-changes observed in this batch.

### Global iCLIP data analysis

Sequencing reads were first demultiplexed using Ultraplex version 1.2.9 (https://github.com/ulelab/ultraplex)[68] and then processed with the nf-core/clipseq version 1.0.0[69] (https://nf-co.re/clipseq/1.0.0). We used BED files with crosslink positions and scores for all analyses. To normalise crosslinks by expression and correct for length for each transcript, we calculated crosslink density (CPM per kb) for exons and divided this by expression (semi-extractability TPM values obtained with Salmon).

### RIC-seq data processing

We trimmed sequencing adapters using Cutadapt,[70] then paired reads were merged with BBMerge.[63] The merged FASTQ files were used as input for Tosca v1.0.0[62] to identify and analyze hybrid reads formed through RNA proximity ligation. To normalise RIC-seq gene-level counts to expression, we used TPM values calculated from RIC-seq nonhybrid reads summarised at the gene-level including all features (exons and introns) with featureCounts.[71] To visualise the regions identified in the hybrid reads, we used the circlize R package.[80]

### RIC-seq network inference and analysis

For each stage, we pooled the RIC-seq deduplicated hybrids files produced by Tosca, filtered out rRNA, tRNA and mitochondrial RNA containing hybrids, and retained the hybrid reads mapping to two different genes (representing intermolecular interactions), and analyzed it using igraph[72,73] version 2.0.3 in R 4.4.0. To assess the connectivity of the smOOPs subgraph, we compared the observed subgraph with control networks generated by degree-matched sub-sampling. For the degree-matched sampling method, we sub-sampled random subnetworks from the full RIC-seq network maintaining the same number of nodes as the number of smOOPs identified in the RIC-seq data at each stage and a similar degree distribution (allowing for ±1° variation). We performed 10,000 iterations to build a null distribution for comparison. Connectivity metrics were calculated for each random subset and compared to those of the observed smOOPs subnetworks for nPSCs and pPSC separately. While we also considered degree-preserving randomization methods, they were deemed less suitable for this analysis due to the network topology and the high degrees of the smOOPs, which could bias the results by preserving inherent connectivity patterns, especially among high-degree nodes.[86] We

also considered random sub-sampling, but found it overestimated smOOPs interconnectivity due to their high degree relative to most other nodes. In contrast, degree-matched sub-sampling, which directly compares smOOPs to randomly selected groups of nodes with similar degrees, provided a more appropriate baseline for comparison, and was used as control.

### PARIS data processing

We first trimmed sequencing adapters using Cutadapt and collapsed PCR duplicates with the readCollapse.pl script provided with the original publication[46] (https://github.com/qczhang/icSHAPE). Subsequent processing was performed using Tosca v1.0.0.[62]

### Ribo-seq data processing

Public mESCs ribo-seq data[41] was processed using the riboseq-flow v1.1.1[77] pipeline. We then applied RiboCode[78] using the alignments to predict translated ORFs from ribosome-protected fragments. Translation efficiency measurements were obtained from the Supplementary Tables accompanying the manuscript.[41]

### Processing of RNP granule gene sets

All genes expressed in mouse embryonic stem cells (mESCs) were used as the starting set. Orthologous human genes were identified through BioMart's orthologue mapping table, enabling cross-species comparison. Human transcriptomes associated with specific subcellular compartments were then compiled by extracting supplementary data from previously published studies: P-body-enriched transcripts,[35] stress granule-associated RNAs,[36] transcripts localized to TIS granules and the endoplasmic reticulum,[37] a subcellular transcriptome atlas generated using APEX proximity labeling[38] and localization of RNA (LoRNA) data obtained by quantifying RNA abundance across density-separated cellular fractions and inferring subcellular identity based on RNA co-distribution profiles.[39] To assess the overlap of smOOPs with RNP granule-associated RNAs, we intersected the gene IDs of smOOPs recovered at all stages with mouse-mapped RNP granule gene sets from each study and visualised the intersections using the ComplexUpset package in R. For enrichment testing, we retained granule sets in which at least 10 smOOPs were exclusively annotated to that compartment (i.e., not overlapping any other granule set). One-sided hypergeometric tests were performed for each retained set, using all genes expressed in our dataset (i.e., the input genes for DESeq2) as the background population.

### lncRNAnet prediction of long non-coding RNA

To determine whether genes annotated as "To Be Experimentally Confirmed" (TEC) are predicted to encode long non-coding RNAs, we first collected their processed sequences into a FASTA file formatted per lncRNAnet[40] instructions (https://github.com/nofundamental/lncRNAnet). These sequences were then processed through the lncRNAnet prediction model consisting of recurrent neural networks modeling RNA sequence features and convolutional neural networks scanning for stop codons to generate an open reading frame indicator, with their outputs being integrated to assign each transcript a continuous confidence score from $-1$ (strongly coding) to $+1$ (strongly non-coding).

### Data preparation for deep learning

To define genes that do not exhibit the condensation prones features in all stages, we selected those with padj $>0.01$ and $|LFC| < 1.4$ in each stage and then selected their intersection as the unified control set. To create a reference transcriptome for the smOOPs and control genes, we selected the most highly expressed transcripts, based on semi-extractability, for each of the three stages. For transcripts that varied across the stages, we selected the longest isoform among the most expressed transcripts at each stage.

To facilitate the training of the model on the selected datasets mapped across each transcript, we first encoded the data into a format suitable for deep learning. For both smOOPs and control transcripts, we extracted the genomic location of each exon from the reference genome. The RNA sequence for each exon was obtained based on their genomic coordinates using bedtools,[74] and served as the raw sequence input for the model.

Several nucleotide-resolution transcriptomic datasets were extracted and aligned to the exon genomic locations: global iCLIP crosslinks (this work); POSTAR3 database of CLIP binding peaks for 46 RBPs[48] in mouse cell lines; Psoralen Analysis of RNA Interactions and Structures (PARIS)[46]; RNAfold[47] predicting secondary RNA structures; and m$^6$A methylation profiles.[45] To precisely assign scores to exon positions, we generated a list of values corresponding to the length of each exon, initialised to zero. The experimental data were mapped onto the exon's genomic coordinates relative to its start position. Overlaps between experimental signals and exonic regions were located, and their scores were added to the list at the corresponding positions. The result was a list of experimental values corresponding to each position within the exon. For POSTAR3 peaks, each RBP peak was mapped in a binary manner, indicating only the presence or absence of a peak. After assigning scores, exons were concatenated back into transcripts, producing a list of feature scores spanning the entire transcript length for each feature. The global iCLIP, m$^6$A modification sites and PARIS-Intra/Inter counts at each nucleotide were first normalised to transcript-level expression (TPM) estimated using the semi-extractability assay data, before encoding all feature layers of information over each smOOPs and control transcript. This approach preserves the positional importance and allows efficient feature extraction. Additionally, RNAfold was used to predict secondary RNA structures in dot-bracket notation, classifying each position as either single- or double-stranded. This classification was then encoded in binary form.

The datasets were randomly split into distinct training, validation, and testing subsets in a 70:15:15 ratio. Stratified sampling was employed to maintain consistent class proportions across subsets, and oversampling was applied to the minority group within each subset to balance class sizes. Each subset was stored independently and used directly for training, with dynamic data fetching and encoding implemented during the training process.

### Data encoding

During training, batches of data were procedurally encoded by iterating over the training dataset and processing the mapped information to a DL compatible array. For each batch, the transcript sequence information was first converted from its native nucleotide string format into a numerical representation. To ensure transparency and interpretability, we employed one-hot encoding for the sequence data, with each nucleotide represented as a binary vector: Adenine (A) as [1, 0, 0, 0], Cytosine (C) as [0, 1, 0, 0], Guanine (G) as [0, 0, 1, 0], and Thymine (T) as [0, 0, 0, 1]. Non-sequence features, such as global iCLIP crosslink scores, were encoded as one-dimensional arrays of floats. All encoded data layers, including the nucleotide sequences and other transcriptomic features, were stacked vertically for each transcript. To then standardise input lengths for neural network processing, sequences were padded with $-1$ to match the length of the globally longest transcript. The output class was similarly encoded using a one-hot encoding scheme, where class membership was represented by a binary vector. Such encoded transcripts were then stacked to a new dimension representing the batch and as such processed by the training function.

### Deep learning model architecture

The model architecture employs a previously optimised design,[45] combining multiple convolutional, recurrent, and fully connected layers (MLP) to effectively classify sequential data, implemented in TensorFlow.[75] This hybrid approach leverages the strengths of each layer type to capture diverse patterns in the input data, progressively transforming the sequences into representations suited for binary classification. Initially, we wanted to enable the architecture of the model to have the ability to achieve optimal classification performance, therefore a hyperparameter optimization process was conducted - defining the models architecture. Due to the complexity and size of the hyperparameter search space, a manual approach was impractical. Instead, a systematic search via Bayesian optimization, using the Optuna framework,[76] was implemented, exploring batch sizes (8, 16, 32, 64), number of convolutional blocks (2–8), final number of CNN units (16–1024), CNN unit increase percentages (0.0–0.5), kernel sizes (3, 5, 7, 9), kernel size increases (1–4), dilation increases (1–4), dropout probabilities (0.0–0.4), L1-L2 regularisation values (0–0.01), max pooling size (2, 3, 4), GRU units (16–1024), Dense units (16–1024), learning rates ($10^{-5}$ to $10^{-3}$), and normalisation methods (None, BatchNormalization, LayerNormalization). The optimisation was performed over 100 trials to identify the hyperparameter set that maximised AUROC on the validation set. The best-performing configuration was selected for final model training. The optimised model architecture starts with four convolutional blocks. Each block consists of a 1D convolutional layer with a progressively increasing number of filters of 108, 144, 192, and 256 respectively, and kernel size of 9. Dilation rates exponentially increase with each block, with the first block using a dilation of 1 and later blocks using 4, 16, and 64, respectively. Each convolutional layer is followed by layer normalisation, a ReLU activation function, and a Dropout layer with a rate of 0.34 to mitigate overfitting. Max-pooling (pool size of 4) is used after each convolutional block to reduce spatial dimensions and preserve key features. After the convolutional layers, a bidirectional GRU layer with 128 units per direction is employed to capture sequence dependencies, again followed by layer normalisation and dropout.

Following the GRU, two fully connected (Dense) layers are included, with 64 and 32 units, respectively, followed by layer normalisation and dropout. The final output layer consists of two units, with a softmax activation to predict binary classes. The Convolutional, GRU and Dense layers were configured with L1-L2 regularisation ($\lambda = 1.2 \times 10^{-5}$) to further mitigate overfitting in the learning process. The model contains between 1,149,986 and 1,203,446 trainable parameters, depending on the input shape. It was trained using the Adam optimizer, with a learning rate optimised to $10^{-4}$, and categorical cross-entropy as the loss function. The model was trained for unlimited epochs, employing early stopping with 20 epoch patience, based on validation AUROC, to stop training and return best weights when overfitting.

### Training of the powerset

To evaluate the predictive power of individual features and their combinations, we trained a separate model for each of the 127 subsets in the powerset of the 7 available datasets. For each subset, only the selected features were encoded as input to the model, ensuring that the model was trained specifically with that particular feature combination. Each model was trained in 8 replicates to account for variability and to ensure optimal performance for every subset. The trained models were saved after completion to enable further analysis of feature importance and interactions.

### Assessing model performance

The final performance of each of the 1016 trained models (127 subsets trained in 8 replicates) was established by predicting the combined validation and testing dataset, both of which were not previously seen by the model, ensuring robust validation and mitigating the optimisation overfitting. Of the 8 replicates trained on each subset, the best performing model, based on the AUROC, was selected, ensuring the robustness of the predictive power assessment. To identify the datasets that improved the predictive power of a model trained on any given subset of datasets, the difference in AUROC was calculated for all possible pairs of models where they were both trained on the same subset of datasets with the only difference being an additional inclusion of the unique dataset.

**Cell Genomics**
**Article**

This was further repeated for all possible pairs for each of the unique datasets and the differences were evaluated using a one-sample t-test to evaluate if the mean of differences is statistically different from zero, Using this technique, we could identify the datasets containing biological information that significantly improved the predictive capability of a model, irrespective of the number of other information layers included. To further assess the informational overlap between features, we conducted pairwise analyses of feature combinations. For each pair of features, we calculated AUROC scores for each subset of each item in their powerset, first by adding one feature from the pair, and then by adding both features. This approach allowed us to quantify the individual and combined contributions of each feature to model performance. To evaluate the extent of overlap, we calculated the difference between the maximum AUROC achieved when adding either feature individually and the AUROC obtained when both features were included. These differences were calculated across all subsets of the remaining five features, resulting in a comprehensive set of values for each feature pair. Averaging these differences provided insight into the distinct contributions of each feature combination with a one-sample t-test comparing the mean to zero.

To validate model performance on the smOOPs identified in the second batch, we constructed a new set of smOOPs and matched control transcripts using the same criteria described above without excluding the smOOPs defined in the previous batch (161 overlapping genes). These samples were processed through the same pipeline, with identical feature encoding, and their class identity was predicted using three models: a length-controlled model, a sequence-only model, and a model incorporating all features.

### Explaining the trained weights

To evaluate the contribution of individual positions and feature components to the model's predictions, we employed the integrated gradients (IG) method. The technique quantifies the importance of each nucleotide in the sequence by computing the gradient of the prediction output with respect to the input features, integrated along a path from a baseline of zeros to the actual input sequence. By summing these gradients across the entire path, IGs assign attribution/importance scores to each nucleotide, indicating its contribution to the prediction. We calculated the IGs for each of the models trained on individual datasets of sequence, global iCLIP, POSTAR3 peaks and PARIS-Intra. This approach enabled us to identify the regions within each transcript that had the greatest impact on the model's classification decision, providing insights into which specific sequence elements were driving the condensation-prone smOOPs RNA predictions compared to controls.

### Gaining insight into the learned features

To gain insight into the global features the model has learned, not limited to a single example but integrated for the entire groups of transcripts, we firstly stacked the integrated gradients of each of the top 4 models to obtain the importance scores over all the predictive features for each transcript. These scores were grouped into 100 bins along the transcript length, and the average importance score within each bin was calculated to create a length-standardised distribution of importance across the sequence. We used Uniform Manifold Approximation and Projection (UMAP)[87] to reduce the binned transcripts to 2D space, using correlation as the distance metric. Agglomerative clustering was then applied to group the transcripts into 2 clusters.

To elucidate the features defining each cluster we continued with the following analysis separately for each. We calculated the importance of nucleotide triplets across the transcripts. For each transcript, a sliding window approach was used to capture triplet sequences, and their integrated gradient scores were averaged for each triplet at every position. This reduced the transcript length by two. The triplet importance scores were then divided into 100 equal-length segments, and the scores were averaged across all transcripts. To further analyze the predictive importance of each POSTAR3 track, we calculated the average importance scores across all bins for each POSTAR3 RBP binding site in each transcript.

### Validation of the predicted importance

To assess nucleotide frequency differences between smOOPs and control transcripts in both clusters, we binned the sequences and averaged the presence of each nucleotide (A, C, G, T) across bins. For cluster 1, the sequence was separated into transcript regions, with the 5′UTR divided into 10 bins and the CDS and 3′UTR each divided into 50 bins. For cluster 2, the entire transcript was divided into 100 bins. The average nucleotide frequency within each bin was calculated separately for smOOPs and control transcripts. The differences in nucleotide frequency were determined by subtracting the control average from the smOOPs average for each nucleotide in each bin. To further investigate sequence composition, we calculated the frequency of each possible nucleotide triplet (3-mer) across the entire sequence for cluster 2 and within the transcript regions for cluster 1. The triplet occurrence frequency for control transcripts was subtracted from that of smOOPs transcripts for each cluster, yielding the difference in triplet usage. A similar approach was applied to the global iCLIP crosslinking signals. For cluster 1, the normalised signal was binned and averaged by transcript regions, with the 5′UTR divided into 10 bins and the CDS and 3′UTR into 50 bins each. For cluster 2, the signal was binned and averaged across the entire transcript into 100 bins. The median signal value for each bin was calculated for both smOOPs and control transcripts, and a 95% confidence interval for the median was estimated using bootstrapping with 1000 samples.

For cluster 1, we evaluated the impact of specific transcript regions on model predictions by masking either sequence, global iCLIP, POSTAR, and PARIS intramolecular interactions features. The entire region of the transcript was masked using zeros for 5′UTR, CDS, and 3′UTR individually. Models trained on each dataset were then used to predict outcomes based on these masked inputs. The AUROC curves were used to identify the region with the highest impact on the prediction when masked. For cluster 1

and 2, the number of PARIS intramolecular hybrids was calculated for each transcript part (5′UTR, CDS, 3′UTR) and full transcripts, respectively. These counts were normalised by transcript region length and expression levels.

### Investigation of the protein features

To generalise the two-cluster annotation obtained in naive PSCs (nPSCs) and apply it to all smOOPs transcripts, we trained an additional machine-learning (ML) model focused solely on RNA sequence features. As before, we used eight replicate training runs and retained the model achieving the highest area under the receiver operating characteristic curve (AUROC). We calculated the importance scores for each nucleotide position via integrated gradients, binned these scores, and clustered all smOOPs based on these aggregate importance profiles. We then evaluated how well the newly defined clusters (across all developmental stages) overlapped with the nPSC-specific clusters by measuring their percentage of intersection.

Next, to examine whether the two smOOPs clusters display distinct protein-level characteristics, we calculated amino acid frequencies for all proteins encoded by transcripts in each cluster and compared them against non-smOOPs. For each amino acid, we computed the $\log_2$fold change in mean usage relative to non-smOOPs and determined statistical significance using Welch's t-test.

We further assessed the extent of protein disorder in each cluster by extracting annotated intrinsically disordered regions (IDRs) from the UniProt Reviewed database.[57] For each protein, we recorded whether it contained any IDRs and calculated the average proportion of the protein length exhibiting disorder. Additionally, we used PICNIC,[52] a deep learning approach that leverages both sequence- and structure-derived information from AlphaFold2 models to evaluate how likely these proteins are to localize in biomolecular condensates.

Lastly, we performed gene ontology (GO) enrichment analysis to elucidate the functional distinctions of the two smOOPs clusters relative to the set of all expressed genes used as input to DESeq2. Using the STRING database,[58] we identified significantly enriched GO terms in each cluster across biological process, molecular function, and cellular component ontologies.

