## [Document S2. Transparent peer review records for Klobučar et al. · Cell Genomics]

Integrative profiling of condensation-prone RNAs during early development

Author list: Tajda Klobučar, Jona Novljan, Ira A. Iosub, Boštjan Kokot, Iztok Urbančič, D. Marc Jones, Anob M. Chakrabarti, Nicholas M. Luscombe, Jernej Ule, Miha Modic

Summary

Initial submission:	Received : February 4 th 2025
	Scientific editor: Judith Nicholson
First round of review:	Number of reviewers: 3
	Revision invited : March 4 th 2025
	Revision received : August 5 th 2025
Second round of review:	Number of reviewers: 3
	Accepted : 20 th October 2025
Data freely available:	Yes
Code freely available:	Yes

This transparent peer review record is not systematically proofread, type-set, or edited. Special characters, formatting, and equations may fail to render properly. Standard procedural text within the editor's letters has been deleted for the sake of brevity, but all official correspondence specific to the manuscript has been preserved.

Referees' reports, first round of review

Reviewer 1:

Comments enter in this field will be shared with the author; your identity will remain anonymous. Klobucar, Novljan, Iosub et al. demonstrate that the overlapping set of semi-extractable and orthogonal organic phase RNAs (after cross-linking to proteins) represent a set of RNAs with distinct chemical behavior. The manuscript then i) argues that these RNAs (smOOPs RNAs) enriches for transcripts associated with RNP granules, and ii) defines the features of these RNAs by computational methods. Overall, the findings in this manuscript are of interest.

The key issue with this manuscript is demonstrating the smOOPs RNAs are enriched in RNAs associated with granules. If this issue can be more rigorously demonstrated, the manuscript would be of importance and appropriate for publication in Cell Genomics. Additional comments to improve the work are detailed below.

This review is from Dylan Parker and Roy Parker, and we would be willing to clarify these comments for the authors if needed.

The Crux of the manuscript:

1) The crux of this manuscript is whether the smOOPs RNAs are enriched in RNAs localized to RNP granules. The presented data addressing this issue is that a subset of the smOOPs RNAs show larger foci in HCR-FISH than a subset of non-smOOPs RNA. However, the chain reaction steps of HCR-FISH make it challenging to ascertain whether specific transcripts are in granules or simply amplify better resulting in larger foci. We suggest the following additional experiments/analyses to determine if smOOPs RNAs define a class of RNAs enriched in RNP granules.

Additional Analyses on existing data.

A) In the HCR-FISH, analyze and show in the supplemental data the intensity distribution of FISH signal for each mRNA. If a smOOPs RNA is in a multimeric

homotypic granule, there should be a distribution of quantal intensities of 1, 2, 3, etc, with a subset of the RNA being unimodal. If the smOOPs RNA simply has better HCR-FISH, then the distribution should be gaussian but around a higher intensity than a non-smOOPs RNA.

(A2) The authors should also note that a smOOPs, or non-smOOPs, RNA could be in a multimeric heterotypic assembly and not show an increased homotypic HCR-FISH signal.

B) In the RIC-Seq analyses, it would be good to show i) the entire cross-linking interaction map of all RNAs, and then ii) highlight the smOOPs RNAs within that network. This would illustrate if the smOOPs RNAs form their own localized network of interactions or are simply interacting with whole transcriptome.

(B2) It would also be of interest to determine if the two clusters of smOOPs RNAs defined by their sequence properties show more within cluster interactions or not.

Additional Experiments.

C) The authors could examine how the distribution of smOOPs RNAs maps onto known RNP granules either by comparing their datasets to constitutive RNP granules of known RNAs (Cajal bodies, Nucleolus, P-bodies) and determining if there is an over-representation in those assemblies.

D) A more direct way to compare smOOPs RNAs to RNP granule assemblies would be perform the smOOPs analysis on a cell line during stress where stress granules form (since the transcriptome of stress granules is well established (Khong et al., 2017; Zhou et al., 2024)). This might be a very important experiment and would also establish if traditional RNA extraction methods miss RNP granule RNAs.

Additional Comments:

2) It is important to describe the criteria for calling a smOOPs RNAs in the main text and not just in the methods. This is central to the logical flow, and I would recommend inclusion in main text.

3) The number of biological replicates for the sequencing or HCR-FISH experiments is not immediately apparent in the text, legends, or figures. These should be provided.

4) A table identifying the list of transcripts examined by HCR-FISH should be included, which could be in the supplemental with each transcript's intensity distribution.

5) It would be useful to see the error bars for each individual transcript point in the aggregated HCR-FISH data to allow an assessment of the robustness of this data.

6) It is common in the field to state that RIC-seq queries RNA-RNA interactions directly, but this isn't necessarily true. RIC-seq detects RNAs that are in proximity due to being in a complex with an RBP. We feel that as the field is moving towards characterizing RNA-RNA interactions within granules the distinction between direct and indirect interactions is an important one. I suggest clarifying this issue in the text.

7) Comparing the smOOPs dataset to mRNAs previously suggested to be in higher order assemblies might help strengthen the conclusion they represent RNP granule components (DOI: 10.1016/j.devcel.2020.07.010).

8) It took a moment to discern what the colors in figure 2D were referencing. A colon following "Null network type" would make this more immediately readable.

9) Since there is a large fraction of mRNAs present in cluster 2, it would be useful to see a gene body representation (start/stop) for the coding subset of this cluster in figure 4 like how cluster 1 is presented in 4B.

9a) Similarly, it is stated that CLIP signal is distributed in cluster 2, but if the scale of the bins is similar for mRNAs in cluster 1 it looks like the CLIP signal may be enriched in the UTRs. Is there an enrichment in any transcript region or is it truly uniformly distributed?

10) In Figure 4F and G, the Y-axes should be the same to illustrate the difference in magnitude that is highlighted in the text.

Reviewer 2:

Comments to Author:

In this study, Klobučar et al. identifies a novel class of RNAs termed smOOPs (semi-extractable, orthogonal organic phase separation-enriched RNAs) that are highly enriched in RNA-binding proteins (RBPs) and exhibit properties consistent with condensation-prone RNAs. These RNAs are localized to larger intracellular foci and form dense RNA-RNA interaction subnetworks. Using RNA in situ conformation sequencing (RIC-seq), the study demonstrates that smOOPs form highly interconnected RNA-RNA interaction (RRI) networks, suggesting their spatial organization within cells. These networks are more densely connected than expected by chance, indicating a functional role in RNA condensation. The authors also employ an explainable deep learning framework to predict smOOPs based on intrinsic RNA features (e.g., sequence composition, RBP binding patterns, and intramolecular folding). The model reveals that smOOPs have lower sequence complexity, increased intramolecular folding, and specific RBP binding patterns. Interestingly, smOOPs encode proteins with extensive intrinsically disordered regions (IDRs) and are predicted to be involved in biomolecular condensates. The findings together suggest that smOOPs play a role in coordinating post-transcriptional gene regulation and may contribute to the formation of biomolecular condensates, which are implicated in various cellular processes and diseases. Overall, this is an important contribution to the RNP field and I recommend publication once the authors address the comment below.

1. While the study provides strong computational and biochemical evidence for the condensation-prone nature of smOOPs, functional validation through experimental approaches (e.g., knockdown, overexpression, or live-cell imaging) would strengthen the conclusions. For example, demonstrating the impact of smOOPs on condensate formation or cellular processes would provide direct evidence of their biological relevance.
2. In Figure 1D, the x-axis label for 'protein-coding' appears to be misaligned. Please adjust the label to its correct position for clarity.

Reviewer 3:

In this manuscript Klobučar and colleagues describe a transcriptomics assay to identify condensation-prone RNAs (smOOPs). These transcripts localize to large intracellular foci, form denser RNA-RNA interaction subnetworks than expected,

and are extensively bound by RNA-binding proteins (RBPs). Through an explainable DL framework, the authors uncover that smOOPs possess distinct sequence features, including reduced complexity, enhanced intramolecular folding, and specific RBP binding patterns. Notably, these RNAs encode proteins with extensive intrinsically disordered regions and are strongly predicted to participate in biomolecular condensates, highlighting the interplay between RNA- and protein-based features in phase separation.

This study is quite intriguing and certainly of general interest. It advances our understanding of condensation-prone RNAs and provides a valuable resource for exploring RNA-driven condensation mechanisms. Before this work should be considered for publication a few issues should be addressed. Code and sequencing information are well documented.

1. The authors argue that their findings suggest that smOOPs are a unique class of semi-extractable transcripts that are highly bound by RBPs and form larger foci within cells expression level smOOPS RNAs versus extraction. Are these foci distinct from nuclear and cytosolic bodies or smaller and occur all over respective compartment? Co-staining with specific cellular maker would help to explain this issue.
2. Are mRNA transcripts with an encoded signal peptide and/or transmembrane domain excluded from the two identified clusters? Meaning ER-targeted transcript are not condensation-prone.
3. Is there any evidence that smOOPs lncRNAs are being translated ? Analyzing mESC ribo-seq data for novel ORFs would provide an answer to this question
4. In Fig4E the authors show that longer transcript are enriched ? Which specific mRNA compartments show this enrichment? Long 5'UTR, CDS, or 3'UTR.
5. There are a number of RNA occupancy profiles published in different cell lines. Do the smOOPs transcript also show increased RBP binding also in other cell types?
6. The RNA occupancy profile of cluster 1 is biased, since cluster 1 transcripts are mostly composed of mRNAs. Elongating ribosomes likely displace RBPs and reduce RBP-RNA interactions. Are these CDSs more bound by RBPs than in non-smOOPs? This would have implication why these transcripts condensate. The authors should repeat the global iCLIP analysis in the presence of ATP depletion or cycloheximide, a strong translation repressor, to assess binding of RBPs in absence of translation. The authors found that the global iCLIP and POSTAR3 were most informative only in the 3'UTR. This might be again due to ribosome

elongation and RBP displacement.

6. The authors should cross-reference their smOOPs transcript, separated in cluster 1 and 2 transcripts, with various datasets (e.g PMID: 24582499, 35020268). In Fig S4G+H, they use single ribo-seq and RNA half-life datasets to compare to all smOOPs. What about cluster 1 and 2 ? How do other mESC ribo-seq and half-life data relate? TE might not a good measure, since ribosome stalling is not captured and could result in increased TE. Does ribosome stalling distinguish smOOPs from non-smOOPs transcripts? There are data now on mRNA cellular flow (PMID: 38753883, 38964322, 39548324)? Are there any differences detectable for kinetic parameters of smOOPs vs non-smOOPs transcripts?

7. The authors describe, the ratio of C-rich to A/U-rich smOOPs shifts during development. In nPSCs, C-rich smOOPs were predominant, comprising approximately 75% of nPSCs-specific smOOPs. In dPSCs, their proportion declined to 56%, reflecting an increased representation of A/U-rich smOOPs at later developmental stages. Why is this the case? Does the cell cycle length or cell cycle distribution at different stages play a role? Past work showed that tRNA expression profile in proliferating and differentiated cells influences codon biases (PMID: 25215487). Could this have an impact here?

Authors' response to the first round of review

Point-by-point response to reviewers' comments:

Reviewer #1: Comments enter in this field will be shared with the author; your identity will remain anonymous. Klobucar, Novljan, Iosub et al. demonstrate that the overlapping set of semi-extractable and orthogonal organic phase RNAs (after cross-linking to proteins) represent a set of RNAs with distinct chemical behavior. The manuscript then i) argues that these RNAs (smOOPs RNAs) enriches for transcripts associated with RNP granules, and ii) defines the features of these RNAs by computational methods. Overall, the findings in this manuscript are of interest.

The key issue with this manuscript is demonstrating the smOOPs RNAs are enriched in RNAs associated with granules. If this issue can be more rigorously demonstrated, the manuscript would be of importance and appropriate for publication in Cell Genomics. Additional comments to improve the work are detailed below. This review is from Dylan Parker and Roy Parker, and we would be willing to clarify these comments for the authors if needed.

The Crux of the manuscript:

1) The crux of this manuscript is whether the smOOPs RNAs are enriched in RNAs localized to RNP granules. The presented data addressing this issue is that a subset of the smOOPs RNAs show larger foci in HCR-FISH than a subset of non-smOOPs RNA. However, the chain reaction steps of HCR-FISH make it challenging to ascertain whether specific transcripts are in granules or simply amplify better resulting in larger foci. We suggest the following additional experiments/analyses to determine if smOOPs RNAs define a class of RNAs enriched in RNP granules.

Additional Analyses on existing data.

A) In the HCR-FISH, analyze and show in the supplemental data the intensity distribution of FISH signal for each mRNA. If a smOOPs RNA is in a multimeric homotypic granule, there should be a distribution of quantal intensities of 1, 2, 3, etc, with a subset of the RNA being unimodal. If the smOOPs RNA simply has better HCR-FISH, then the distribution should be gaussian but around a higher intensity than a non-smOOPs RNA.

We thank the reviewers for the mechanistically grounded suggestion to examine the intensity distribution of HCR-FISH signal for each mRNA. We have now analysed intensity distribution per foci of the targeted RNAs as displayed in Figure R1. While we do not observe discrete quantal intensities that would be expected from well-separated multimers, the intensity distribution for smOOPs RNAs displayed visibly non-Gaussian distribution with long right tails in the density plots (Figure R1). This was further supported by performing Lilliefors (Kolmogorov-Smirnov) normality test, which confirmed that the

intensity distribution for both smOOPs and non-smOOPs is not normal (p -value $< 2.2e-16$).

However, we would like to clarify that our study does not propose that smOOPs form multimeric homotypic assemblies, as would be required to observe clear quantal intensity distribution in multimodal peaks (e.g. 1x, 2x, 3x). This is only one of the several possibilities how these RNAs cluster and we now discuss this more clearly in the text and the response to A2, where you correctly pointed out that larger foci by HCR-FISH do not necessarily imply that these assemblies are homotypic. Furthermore, we believe that detecting stepwise quantal distribution is technically infeasible with the hybridisation-chain-reaction-FISH using confocal-level microscopy due to the nonlinear nature of signal amplification (Choi et al. 2018). Hybridisation-chain-reaction is leading to a rather stochastic amplification of signal per RNA, meaning that there is a wide distribution of what intensity level can be obtained, very different from a single fluorophore - consistent smFISH labeling approach. As pointed out by (Trcek et al. 2020), resolving true molecular stoichiometry would likely require super-resolution imaging in conjunction with single-molecule FISH (smFISH) approach.

Figure R1: The intensity distribution of HCR-FISH signal for each mRNA ($n=17$), separated by fluorophore (full line is shows the distribution of mean intensity of individual foci, measured for each smOOPs, dashed line of non-smOOPs; the n is showing the number of foci detected per transcript).

To mitigate potential amplification-related artifacts, we implemented further technical controls. As individual fluorophores might behave differently, we used three different amplifier fluorophores (488nm, 594nm and 640nm), applied across smOOPs and non-smOOPs, to ensure that colour-specific fluorophore properties did not bias signal strength. This way we could eliminate the possibility that some fluorophores are brighter.

We would also like to emphasize that similar DYNC1H1 mRNA clustering behavior was reported by (Pichon et al. 2016), who used 36 classical "Stellaris-type" DNA probes to detect DYNC1H1 mRNA in HeLa cells. Notably, both the published smFISH approach utilizing conventional Stellaris probes and our HCR-FISH approach produce consistent observations. This concordance strongly supports the conclusion that the increased

fluorescence signal arises from mRNA clustering, rather than from uneven signal amplification during HCR labeling.

We hope these analyses and technical clarifications address the reviewers' question. While we recognise the limitations of amplification-assisted FISH methods, the similar intensity distributions of all RNAs in question, among which we could also reproduce the known clustering behaviour of *Dync1h1*, led us to believe that our approach provides biologically meaningful insights and the observed larger foci formed by smOOPs are not a result of any technical artefact.

(A2) The authors should also note that a smOOPs, or non-smOOPs, RNA could be in a multimeric heterotypic assembly and not show an increased homotypic HCR-FISH signal. We thank the reviewers for the comment and we fully agree that the initial writing has overlooked this possibility. Indeed, our observation that smOOPs RNAs tend to form larger foci by HCR-FISH does not necessarily imply that these assemblies are homotypic. As the reviewer rightly points out, such foci could also reflect multimeric heterotypic assemblies involving multiple RNA species that co-condense and jointly occupy a larger subcellular volume. Addressing this question would require multiplexed imaging approaches capable of resolving the spatial co-occurrence of multiple transcripts at single-molecule resolution. Towards this goal, we are currently preparing a MERFISH imaging experiment using the Merscope Ultra platform, which will enable sequential profiling of smOOPs RNAs *in situ* and provide insights into their organization within potentially larger, heterogeneous condensates. We envision this experiment forming the basis of a follow-up study focused on the higher-order spatial logic of smOOPs organization and condensate co-occupancy. However, we believe that this is beyond the scope of the current manuscript that is rich in classifying many features that smOOPs RNAs exert.

To this end, we have added this sentence in the second section of Results:

"While these are clearly visible foci of a single RNA, it does not exclude the possibility of them being a part of heterotypic assemblies that consequently occupy a larger subcellular area."

B) In the RIC-Seq analyses, it would be good to show i) the entire cross-linking interaction map of all RNAs, and then ii) highlight the smOOPs RNAs within that network. This would illustrate if the smOOPs RNAs form their own localized network of interactions or are simply interacting with whole transcriptome.

We thank the reviewers for the suggestion to display the full RNA-RNA interaction network and highlight the smOOPs RNAs. However, given the very large size and complexity of the RIC-seq networks (hundreds of thousands of detected interactions across thousands of genes), visualisation of the full network, or even the smOOPs-only network (e.g. Figure R2; Figure 2), results in highly dense plots where individual interactions and structures are not discernible. We therefore focused on quantitative network analyses to assess the organisational properties of smOOPs, using connectivity metrics such as degree (number of distinct RNA partners), edge density (how interconnected the RNAs are), largest connected component size (the number of RNAs forming the largest interconnected group), and clustering coefficient (the tendency to form tightly-connected groups).

We agree that it is important to determine whether general connectivity with the transcriptome could explain the observed smOOPs interconnectivity. Indeed, smOOPs are highly connected (i.e., they have high degree), interacting with a large number of transcripts, as shown by their degree distribution (number of different connections, Figure S2D). We show that transcript length and expression levels, which correlate with interaction detection sensitivity in RIC-seq, are major contributors to this increased general connectivity.

Importantly, we performed additional analyses to ensure that the observed higher connectivity among smOOPs is not simply due to their general connectivity within the transcriptome. To do this, we constructed degree-matched random subnetworks as null models for comparison (Figure 2D), by generating control groups of RNAs with a similar number of interaction partners as the smOOPs. These degree-matched subnetworks were then compared to the smOOPs subnetworks, and define the random expectations for connectivity used in our analyses. This ensures that we are comparing smOOPs to other RNAs with the same general level of connectivity, so that any observed differences in how tightly smOOPs interact with each other are not just because they have more interactions overall. Our analyses demonstrate that smOOPs subnetworks consistently show greater internal connectivity than expected, even when controlling for general connectivity with the whole transcriptome, indicating that smOOPs preferentially interact with each other rather than randomly across the transcriptome.

While these results were present in our initial submission, to further clarify these concepts and goals of the analysis, we have revised the relevant Results section to explicitly explain the use of connectivity metrics and the rationale for degree-matched random subnetworks:

"Because nodes with high degree are inherently more likely to connect, we specifically tested whether smOOPs preferentially connect with each other, rather than being broadly or randomly connected across the transcriptome. To do this, we generated degree-matched random subnetworks by sampling RNAs with a similar degree as smOOPs. These degree-matched random subnetworks provide a null expectation for connectivity, enabling comparison to the observed smOOPs subnetwork while controlling for biases related to degree, expression, and transcript length."

"Together, these findings suggest that smOOPs are more interconnected among themselves even when controlling for general connectivity with the whole transcriptome, reflecting a specific network organisation that points to their proximity in cells."

While we recognise the value of visual summaries, in this case quantitative approaches provide a more accurate and interpretable assessment of network organisation than direct visualisation could achieve. We also note that all hybrid interaction data are shared in tabular format as Supplementary Files available at ArrayExpress, allowing readers to explore or visualise the network data independently if desired.

(B2) It would also be of interest to determine if the two clusters of smOOPs RNAs defined by their sequence properties show more within cluster interactions or not.

We thank the reviewer for the suggestion to examine whether the two sequence-defined smOOPs clusters show preferential within-cluster connectivity. To address this, we tested whether the two clusters form distinct "communities" within the smOOPs RNA-RNA interaction network - that is, whether RNAs within each cluster tend to interact more with each other than with RNAs from the other cluster. Thus, we computed modularity for the smOOPs subnetwork using the two sequence-defined clusters as predefined node labels. We used a standard network metric called modularity, which quantifies how well a given group assignment matches the structure of the network: high modularity means most interactions happen within groups, while low modularity suggests little or no group structure.

The resulting modularity scores were very low (nPSCs: -0.005; pPSCs: 0.0003), indicating that the sequence-defined clusters do not correspond to separate or well-connected subgroups in the network. We interpret this as evidence that the clusters do not exhibit preferential within-cluster RNA-RNA interactions - instead, they are well interconnected.

For illustration, we include below network visualisations of the smOOPs subgraphs with nodes coloured by cluster label and arranged using a layout that spatially separates the

two clusters, making between-cluster interactions easier to identify (i.e., longer edges) (Figure R2). Together with the modularity analysis, these visuals indicate that the clusters are intermixed and do not show strong within-group interaction preference. We felt this analysis, while informative, was not central to our main conclusions and was unlikely to significantly strengthen the manuscript. However, we fully agree it was a worthwhile question and have included the analysis and interpretation in our reviewer response.

Figure R2: Network visualisations of the smOOPs subgraphs with nodes coloured by cluster label and arranged using a layout that spatially separates the two clusters.

Additional

Experiments.

C) The authors could examine how the distribution of smOOPs RNAs maps onto known RNP granules either by comparing their datasets to constitutive RNP granules of known RNAs (Cajal bodies, Nucleolus, P-bodies) and determining if there is an over-representation in those assemblies.

We thank the reviewer for this excellent suggestion to examine the overlap between smOOPs RNAs and known RNP granules. We have now conducted an extensive analysis combining orthogonal datasets (produced in human cell lines), imaging experiments, and comparative transcriptomics, with a particular focus on P-bodies, stress granules and other cytosolic compartments.

First, we examined how the distribution of smOOP RNAs maps onto known RNP granules by comparing our dataset to publicly available RNA-seq profiles of constitutive RNP assemblies, including P-bodies (Hubstenberger et al. 2017), stress granules (Khong et al. 2017), TIS granules (Horste et al. 2023), nuclear and cytoplasmic compartments

profiled by APEX-seq (Fazal et al. 2019) and RNP granules obtained by density-based method LoRNA (Villanueva et al. 2024). We observed that smOOPs are enriched in multiple granules, with the strongest over-representation in stress granules (Figure R3 and revised Figure 1D). We further investigated this enrichment and provide a detailed summary in answer D below.

Figure R3 and revised Figure 1C,D: C) Mapping of smOOPs onto known RNP granules/membraneless compartments. D) Enrichment of RNAs within characterised RNP granules in the smOOPs set. Overrepresentation was assessed using a hypergeometric test, using only granules data with at least 10 genes not overlapping other granules.

To independently validate the association of smOOPs with RNP granules, we compared our dataset to LoRNA-profiled RNA association across a wide range of organelles and RNP granules (Villanueva et al. 2024). A higher proportion of smOOPs appeared in the “membrane” and “granule” fractions, relative to the total transcriptome background (Figure R4A). This provides orthogonal support for their enrichment in RNP granules and reinforces the idea that smOOPs are condensation-prone.

Moreover, the enrichment of smOOPs within the “membrane” fraction (which through sedimentation assay combines mitochondrial and ER-localised transcripts) (Villanueva et al. 2024), Figure R4A) was further confirmed by observing a higher proportion of smOOPs within the rough ER-enriched transcripts, extracted from particle sorting-based analysis (Horste et al. 2023) and APEX-seq (Fazal et al. 2019) (Figure R4C). Combined, this corroborates ER-enriched transcripts could represent a subset of smOOPs. Interestingly, recent report has shown that transcripts within the stress granules cores in yeast and HEK293T cells are enriched for ER membrane components (Demeshkina and Ferré-D’Amaré 2025), which could potentially explain the enrichment of smOOPs within both stress granules and ER-linked transcriptomes.

Figure R4: A) Mean proportion of all smOOPs and non-smOOPs (combined data from all three stages: nPSCs, pPSCs and dPSCs) found within different compartments, as identified by gradient-based cell fractionation in (Villanueva et al. 2024) (Mann–Whitney U test). B) Mean proportion of all smOOPs and non-smOOPs found within TIS granules, rough ER and cytoplasm, as identified by particle sorting by (Horste et al. 2023) (Mann–Whitney U test). C) Proportion of all smOOPs and non-smOOPs enriched in transcriptomes of the respective compartments, determined by proximity labeling (Fazal et al. 2019) (Fisher exact test), (* $p < 0.05$, ** $p < 0.01$, *** $p < 0.001$).

To experimentally validate these findings in our cellular model (murine nPSCs), we performed HCR-FISH with DDX6 counterstaining to assess colocalisation of 8 transcripts (6 smOOPs and 2 non-smOOPs) with P-bodies. Of these, only *Nfx1* and *Tjp1* mRNAs showed > 30% of RNA colocalisation with DDX6-defined P-bodies (Figure R5D), consistent with published transcriptomic data (Figure R5B). Importantly, most tested smOOPs showed little colocalisation with P-bodies in both published RNA-seq and HCR-FISH results (Figure R5), supporting the interpretation that while some smOOPs localise to P-bodies, not all of them can be found within P-bodies.

Figure R5: Enrichment of smOOPs within P-bodies. A) Enrichment of all identified smOOPs and non-smOOPs in P-bodies relative to total RNA (log₂FC), as determined by (Hubstenberger et al. 2017). B) Table showing the enrichment (P-body log₂FC) of selected transcripts used for HCR-FISH (from the data obtained in (Hubstenberger et al. 2017)). C) Representative HCR-FISH photomicrographs (scale bar: 5 μm). D) Quantification of HCR-FISH coupled with DDX6 IF staining showing the percentage of RNA signal colocalising with P-bodies for each selected transcript (the number of histograms depicts the number of frames used for quantification, total number of cells counted 50-70).

Combined, these data supported the emerging view that smOOPs do not belong to a single granule type, but are heterogeneously distributed across multiple condensate classes and hence indeed represent a transcriptome subgroup with unique condensation-prone properties.

D) A more direct way to compare smOOPs RNAs to RNP granule assemblies would be to perform the smOOPs analysis on a cell line during stress where stress granules form (since the transcriptome of stress granules is well established (Khong et al., 2017; Zhou et al., 2024)). This might be a very important experiment and would also establish if traditional RNA extraction methods miss RNP granule RNAs.

We thank the reviewer for the interesting suggestion to extend our results and to determine whether our approach would be applicable to the isolation of stress granule RNAs. Overlapping smOOPs with the core stress granule transcriptome (Khong et al. 2017)) showed they are more enriched in stress granules than non-smOOPs (Figure R3; Figure R6A), thus, as the reviewer suggested, SG-inducing conditions are highly relevant for studying smOOPs. Thus, we performed imaging experiments and repeated the semi-extractability assay and OOPS in non-stressed and thapsigargin-treated nPSCs. We structure our answer in three relevant sections:

1. SG-induction optimisation and imaging

To determine the appropriate stress conditions in this cell type, we first tested several stressors and assessed the percentage of cells with stress granules (SG) based on reports in mouse embryonic stem cells (Ries et al. 2019; Pattabiraman et al. 2020; Arimoto-Matsuzaki, Saito, and Takekawa 2016). While sodium arsenite (0.5 mM, 1 hour) led to most cells displaying stress granules, they were extremely big (over 2 μm in diameter). On the other hand, puromycin treatment (1 $\mu\text{g}/\text{ml}$, 2.5 hours) and heat shock (42°C, 30 min) caused the cells to detach completely. Treatment with 10 μM thapsigargin (30 min) led to the 25 - 30% of cells forming stress granules (with median diameter 1.4 μm) but did not cause cells to detach, hence we selected this for imaging and sequencing experiments.

We coupled HCR-FISH with G3BP1 counterstaining for selected smOOPs and non-smOOPs after stress induction (Figure R6B,C) and confirmed that smOOPs on average colocalise more strongly with stress granules than non-smOOPs (Figure R6D). However, we would like to emphasise that enrichment in SG is not exclusive to smOOPs, as certain selected non-smOOPs also displayed noticeable colocalisation with G3BP1 counterstaining, and the measurements across individual transcripts were broadly distributed (Figure R6E). One of the explanations for this could be the previously observed heterogeneous composition of stress granule cores (Demeshkina and Ferré-D'Amaré 2025), however further work specifically on SG induced in nPSCs would be needed to explain our observations.

Figure R6: Enrichment of smOOPs within stress granules. A) Enrichment of all identified smOOPs and non-smOOPs in stress granules relative to total RNA (log₂FC), as determined by (Khong et al. 2017). B) Table showing the enrichment of selected transcripts validated with HCR-FISH (from the data obtained in (Khong et al. 2017)). C) Representative HCR-FISH photomicrographs showing DAPI and G3BP1 (protein marker of stress granules) (scale bar: 10 μ m). D) Quantification of HCR-FISH coupled with G3BP1 IF staining showing the average percentage of RNA signal colocalising with stress granules for the selected smOOPs (n=11) and non-smOOPs (n=5). E) Same as D) but for each selected transcript. For quantification, only cells with stress granules were selected (that was between 18 and 40 cells for each transcript). The blue dots on the violin plot represent a proportion determined per image frame acquired and the black dots represent the mean.

2. Reproducibility of smOOPs identification in nPSCs

We performed additional semi-extractability and OOPS assays in nPSCs under stress and no-stress conditions in three replicates. The no-stress condition is equivalent to the nPSCs in the original manuscript but was performed from a different passage ~4 years later using a slightly different TRIzol formulation.

At the transcriptome level, this 2nd batch correlated well with the first (Figure R7A), though the Control samples showed slightly lower correlation and increased expression in the second batch, including for smOOPs. As Control serves as the reference for differential expression, this shift—likely due to the TRIzol formulation—propagates to fold-

change calculations and smOOPs identification. Despite these technical factors, approximately 30% (166) of smOOPs overlapped exactly between batches (Figure R7B, revised Figure S3E). The distribution of fold-changes showed concordant directional changes for most remaining genes in OOPS and semi-extractability assays, although a subset of the original smOOPs (~120) reversed direction in either assay (Figure R7C). These discrepancies are relevant for the reproducibility of semi-extractability and OOPS assays at the level of individual genes - these methods are sensitive to extraction chemistry, batch effects, and technical drift across long time gaps. This variability was observed in previous reports: semi-extractable RNAs across five human cell lines showed minimal overlap (18 shared genes; (Zeng et al. 2023)).

Crucially, the defining properties of smOOPs were retained: our deep learning model trained on the 1st batch data remained predictive on the 2nd batch (AUROC 0.94 vs.0.79, Figure R7D, revised Figure S3F). This supports that the model captures meaningful, shared smOOP features beyond batch-specific signals. Moreover, the core findings at the protein level were also recapitulated, with the 2nd batch also encoding proteins with a high percentage of IDRs and elevated PICNIC scores (Figure R7E, revised Figure S5E). As this strengthens the notion that it is the specific features that define this class of RNAs, we have now included these results in the manuscript:

"To confirm that the DL model accurately classifies smOOPs using unseen data from a different experimental batch, we repeated the semi-extractability assay and OOPS in nPSCs. The new batch produced a partially overlapping but not identical smOOPs set compared with the original dataset (Figure S3E), which served as an independent test set for model evaluation. We observed a reasonably strong generalisation of the model to the new smOOPs and control datasets with the all features model achieving an AUROC of 0.79 (Figure S3F). Again, the predictive performance of the DL model trained on all features or on the sequence alone (AUROC of 0.75) exceeded the model trained on transcript length alone (AUROC of 0.67), emphasising that the selected features are robust predictors across batches."

We acknowledge the batch variability in the manuscript, which suggests that smOOPs are best interpreted as an assay-defined class with shared features, rather than a fixed set of condensation-prone RNAs. The partial overlap likely reflects both biological dynamics and the technical sensitivity of fractionation-based assays.

"We view smOOPs as an assay-defined subset with shared features, but the precise membership may vary between individual OOPS and semi-extractability isolations due to biological or technical factors."

"Repeating both assays years later in nPSCs revealed that the described class of RNAs (smOOPs) is primarily defined by their features, which remained consistently predictive, even though the specific gene-level identities of smOOPs differed between batches."

3. Comparison of smOOPs in thapsigargin-treated and non-stressed cells

Finally, to directly address the reviewer's question regarding stress, we compared the smOOPs mRNAs in stressed and non-stressed conditions within the 2nd batch. We observed an increased number of smOOPs under stress (Figure R7F). This expansion of the smOOPs class upon stress supports the view that many transcripts could become increasingly condensation-prone, and as such they are extracted to a lesser extent using the standard TRIzol RNA extraction protocol.

As shown in Figure R7G, both conditions yielded significant enrichment for SG-resident RNAs (hypergeometric test, $-\log_{10}p > 25$). The non-stressed smOOPs fraction shows a higher fold enrichment (Observed/Expected $\approx 2.2\times$), while the stressed condition captures a larger absolute number of SG transcripts ($n = 235$ vs. $n = 175$) and reaches greater statistical significance. These results support that smOOPs recover RNAs associated with RNP granules even in the absence of stress, and this capacity increases further during granule formation. Although we cannot establish whether all or most of the smOOPs identified exclusively under stress are localising to SGs, our data supports the idea that conventional RNA extraction methods may underestimate the abundance of these transcripts under both basal and stress conditions.

Figure R7: A) Scatter plots showing the correlation between mean TPM values of genes in control (standard TRIzol), semi-extractability and OOPS RNA-seq libraries between the two batches in nPSCs. B) Venn diagram showing the overlap between nPSCs smOOPs, recovered in the original dataset (first batch) and the second batch. C) Density plots comparing fold changes of semi-extractability and OOPS vs. control for nPSCs smOOPs recovered from either both or single batch. D) AUROC curves showing performance on second-batch smOOP classification. The curves illustrate the true positive rate versus false positive rate

for three models trained on first-batch data: the baseline length-only model (AUROC = 0.667), the sequence-only model (AUROC = 0.750), and the all-features model (AUROC = 0.785). E) Percentage of proteins with IDRs and measures of disorder for nPSCs smOOPs and non-smOOPs, identified from a second batch. Bar chart shows the percentage of proteins with IDRs and the box plots show the percentage of disorder in proteins, their mass and PICNIC score. F) Overlap between smOOPs identified in non-stressed and thapsigargin-treated nPSCs. G) Enrichment of SG-resident RNAs (Khong et al. 2017) in the smOOPs pool in each condition.

These results raise exciting questions about the behaviour of smOOPs under stress, which warrant systematic investigation. For instance, it remains to be determined how smOOPs identities and features shift over the course of stress induction and in response to different stressors (e.g., oxidative, ER, heat). Recent work (Glauninger et al. 2024; Zhou et al. 2024) has shown that the SG transcriptome evolves dynamically, raising the question whether smOOPs may act as early scaffolds or late-stage condensate components. It would also be interesting to explore whether smOOPs contribute to G3BP-stabilised RNA-RNA interaction networks, and whether G3BP is required for their condensation under stress (e.g., via knockdowns and temporal profiling).

While we agree these are compelling avenues and thank the reviewer for highlighting them, we feel such stress-focused studies lie outside the scope of this manuscript, which centres on RNA condensation under homeostatic, non-stress conditions during developmental transitions. Introducing a single stress-related experiment here may open more mechanistic questions than it answers. Nonetheless, we are happy to include these points in the manuscript if the reviewer deems it essential.

Additional

Comments:

2) It is important to describe the criteria for calling a smOOPs RNAs in the main text and not just in the methods. This is central to the logical flow, and I would recommend inclusion in the main text.

We appreciate the reviewer's comment regarding the text flow. Upon reviewing the section in question, we respectfully note that the criteria for defining smOOPs is present in the manuscript in the second paragraph of the Results section (printed below).

"We next performed differential expression analysis to identify semi-extractable and OOPS-enriched genes compared to standard TRIZOL RNA-seq controls at each developmental stage (Figure S1C, Figure 1B, Data S1-S6). By employing stringent effect-size and statistical cutoffs (at least two-fold enrichment and adjusted p-value < 0.01), we identified 449, 1328 and 1390 high-confidence genes at each developmental stage with distinctly increased semi-extractability and elevated RBP occupancy in OOPS samples, henceforth collectively referred to as smOOPs (semi-extractable and OOPS-enriched RNAs)."

For clarity, we have now also included these thresholds in the legend of Figure 1B:

"At each stage, the genes enriched more than two-fold compared to control ($p_{adj} < 0.01$ and $LFC > 1$) in both assays were defined as smOOPs."

3) The number of biological replicates for the sequencing or HCR-FISH experiments is not immediately apparent in the text, legends, or figures. These should be provided.

We appreciate the reviewer's careful reading, which rightly pointed out that information about the replicates was only included in the Methods section. We have now added the number of replicates used in the experiments in the respective figure legends.

4) A table identifying the list of transcripts examined by HCR-FISH should be included, which could be in the supplemental with each transcript's intensity distribution.

We thank the reviewer for this helpful suggestion. In response, we created a new supplementary table (Table S2) listing all transcript targets examined by HCR-FISH and the quantitative measurements used in Figure 1 and Figure S1. To ensure reproducibility, we have also deposited the raw microscopy images and analysis macro files on Zenodo (10.5281/zenodo.13860870). The distribution data for transcript intensities and foci sizes are provided there as a downloadable table instead of embedding it as an additional table, due to file size constraints.

5) It would be useful to see the error bars for each individual transcript point in the aggregated HCR-FISH data to allow an assessment of the robustness of this data.

We thank the reviewer for this useful comment for the improvement of aggregated HCR-FISH data representation. For a better assessment of the data robustness, we have added a box plot showing the size of all RNA foci of individual targets in supplementary figure (revised Figure S1H). We hope you can appreciate that for smOOPs, the distribution of the foci size is more heavy tailed with many extreme values compared to non-smOOPs, despite having counted similar number of foci (mean foci number for smOOPs transcripts = 2698, for non-smOOPs = 2145). In our opinion, this new visualisation enhances the explainability of the data and depicts its spread more than simply adding error bars would in the initial boxplot.

H

Figure R8 and revised Figure S1H: Foci size (area in μm^2) of all foci detected per transcript, analysed by HCR-FISH in nPSCs with boxplots showing the median and interquartile range.

6) It is common in the field to state that RIC-seq queries RNA-RNA interactions directly, but this isn't necessarily true. RIC-seq detects RNAs that are in proximity due to being in a complex with an RBP. We feel that as the field is moving towards characterizing RNA-RNA interactions within granules the distinction between direct and indirect interactions is an important one. I suggest clarifying this issue in the text.

We fully agree that RIC-seq captures RNA-RNA proximity mediated through RNP complexes and does not reflect direct interactions. To address this, we have revised the text in the Results section to explicitly reflect that RIC-seq measures RBP-associated RNA-RNA proximity rather than direct contacts.

"Given the well-established role of RBPs in regulating RNA assembly, we hypothesised that smOOPs participate in broader RNA networks. We applied RNA in situ conformation sequencing (RIC-seq (Cai et al. 2020)) to map intra- and intermolecular RNA proximities across the transcriptome, capturing both direct and indirect, RBP-associated RNA-RNA interactions (RRI) in nPSCs and pPSCs."

We also note that our RIC-seq section conclusion states that the observed smOOPs RNA-RNA connectivity indicates spatial proximity rather than direct interactions: *"Together, these findings suggest that smOOPs are more interconnected among themselves even when controlling for general connectivity with the whole transcriptome, reflecting a specific network organisation that points to their proximity in cells."*

We hope this clarification resolves the concern and appropriately reflects the current understanding of what RIC-seq data can and cannot reveal.

7) Comparing the smOOPs dataset to mRNAs previously suggested to be in higher order assemblies might help strengthen the conclusion they represent RNP granule components (DOI: 10.1016/j.devcel.2020.07.010).

We agree that comparing the smOOPs dataset to mRNAs previously implicated in higher-order assemblies adds strength to our conclusion regarding their potential role as RNP granule components. We have expanded the analysis in response to point C of your review by mapping smOOPs to known RNP condensates (Figure R3, revised Figure 1C), and by assessing overrepresentation (i.e enrichment) of RNAs known to be in granules among smOOPs (Figure R3 and revised Figure 1D). A closer examination of specifically localised human transcripts (Chouaib et al. 2020) showed that 8 of 36 such mRNAs were also smOOPs ($p = 0.0057$; Fisher's Exact test). Together with the observed enrichment of smOOPs in multiple RNP granules (Figure R3), this raises the possibility that a subset of smOOPs may represent locally translated mRNAs. Investigating this in the same cell line (murine nPSCs) would require detailed, transcript-specific analyses beyond the scope of this computational study.

8) It took a moment to discern what the colors in figure 2D were referencing. A colon following "Null network type" would make this more immediately readable.

We thank the reviewer for the suggestion, we have implemented this in the main Figure 2.

9) Since there is a large fraction of mRNAs present in cluster 2, it would be useful to see a gene body representation (start/stop) for the coding subset of this cluster in figure 4 like how cluster 1 is presented in 4B.

We thank the reviewer for raising the question regarding the features of the coding subset within cluster 2. Although this cluster contains only 43 mRNAs, and just 10 meet the minimum 5'UTR, CDS, and 3'UTR length cutoffs of 10, 50, and 50 nt required for binning,

their sequence composition mirrors that of the full cluster, with A/U enrichment observed across both UTRs and CDS.

Figure R9: Per-bin difference in average nucleotide content between the mRNAs in cluster 2 smOOPs (n=10) and control transcripts, with nucleotide content divided into 100 bins across the transcripts.

9a) Similarly, it is stated that CLIP signal is distributed in cluster 2, but if the scale of the bins is similar for mRNAs in cluster 1 it looks like the CLIP signal may be enriched in the UTRs. Is there an enrichment in any transcript region or is it truly uniformly distributed?

We appreciate the reviewer for his thorough reading and detailed question. After visualising the distribution of the global iCLIP signal across the individual regions of the protein coding nPSCs' smOOPs Cluster 2 transcripts (n=10), we can indeed observe a similar enrichment of RBP binding at the 3'UTR. While the actual difference between this group of smOOPs and the control is as large as the difference observed for the total smOOPs from cluster 2, their subset of mRNAs seems to follow the 3'UTR binding preference as observed in the mRNAs from cluster 1. We have also included this in the Results section and included the graph in the revised Figure S4G.

Figure R10 and revised Figure S4G: Median global iCLIP signal, normalised for expression and binned (10, 50, and 50 bins for 5'UTR, CDS and 3'UTR respectively) for protein coding cluster 2 smOOPs and control transcripts. Shaded areas represent 95% confidence intervals per bin, estimated via bootstrapping.

10) In Figure 4F and G, the Y-axes should be the same to illustrate the difference in magnitude that is highlighted in the text.

We thank the reviewer for the well placed comment. In response, we have corrected the figures to reflect the different scales of enrichment in both clusters (revised Figure 4D-G).

Reviewer #2: In this study, Klobučar et al. identifies a novel class of RNAs termed smOOPs (semi-extractable, orthogonal organic phase separation-enriched RNAs) that are highly enriched in RNA-binding proteins (RBPs) and exhibit properties consistent with condensation-prone RNAs. These RNAs are localized to larger intracellular foci and form dense RNA-RNA interaction subnetworks. Using RNA in situ conformation sequencing (RIC-seq), the study demonstrates that smOOPs form highly interconnected RNA-RNA interaction (RRI) networks, suggesting their spatial organization within cells. These networks are more densely connected than expected by chance, indicating a functional role in RNA condensation. The authors also employ an explainable deep learning framework to predict smOOPs based on intrinsic RNA features (e.g., sequence composition, RBP binding patterns, and intramolecular folding). The model reveals that smOOPs have lower sequence complexity, increased intramolecular folding, and specific RBP binding patterns. Interestingly, smOOPs encode proteins with extensive intrinsically disordered regions (IDRs) and are predicted to be involved in biomolecular condensates. The findings together suggest that smOOPs play a role in coordinating post-transcriptional gene regulation and may contribute to the formation of biomolecular condensates, which are implicated in various cellular processes and diseases. Overall, this is an important contribution to the RNP field and I recommend publication once the authors address the comment below.

1. While the study provides strong computational and biochemical evidence for the condensation-prone nature of smOOPs, functional validation through experimental approaches (e.g., knockdown, overexpression, or live-cell imaging) would strengthen the conclusions. For example, demonstrating the impact of smOOPs on condensate formation or cellular processes would provide direct evidence of their biological relevance.

We thank the reviewer for this thoughtful and important comment regarding functional validation of smOOPs. To this end we have now implemented several orthogonal strategies that strengthen the functional interpretation of their condensation-prone nature:

Localisation to known RNP granules:

We now provide an insight into smOOPs intracellular localisation and their enrichment within the transcriptomes of characterised RNP granules (Figure R11A,B and revised Figure 1C,D). Comparisons of smOOPs with orthogonal datasets from transcriptome-wide fractionation assays (Villanueva et al. 2024) and particle-sorting based assays in human cells (Horste et al. 2023; Hubstenberger et al. 2017; Khong et al. 2017) revealed that smOOPs are enriched in various granule-like compartments, including P-bodies (Figure R11C) and stress granules (Figure R11D). We validated these findings experimentally in murine naive pluripotent stem cells (nPSCs) by co-staining smOOPs with granule markers DDX6 (Figure R11D,E) and G3BP1 (Figure R11G,H), showing that while smOOPs are heterogeneously distributed, several preferentially localise to known

RNP compartments. Next, we recapitulated known behaviors of well-established clustering RNAs such as *Dync1h1*, showing consistent results between our HCR-FISH signal and previously published smFISH data (Pichon et al. 2016). This combined reinforces that smOOPs as a group of transcripts indeed exhibit a true subcellular condensation-prone behaviour, and contain many subgroups of transcripts, characterised in specific condensates.

smOOPs upon stress induction:

In line with the suggestion of Reviewer 1(D), we wanted to decipher the involvement of smOOPs on condensate formation. For this we used stress granules as hallmarks of biomolecular condensates and performed stress induction (thapsigargin) in nPSCs. Using HCR-FISH coupled with G3BP1 counterstaining as stress granule marker, we confirmed that smOOPs on average colocalise more strongly with stress granules than non-smOOPs (Figure R11H). However, we would like to emphasise that enrichment in SG is not exclusive to smOOPs, as certain selected non-smOOPs also displayed noticeable colocalisation with G3BP1 counterstaining, and the measurements across individual transcripts were broadly distributed. One of the explanations for this could be previously observed heterogeneous composition of stress granule cores (Demeshkina and Ferré-D'Amaré 2025).

We also repeated the semi-extractability assay and OOPS in non-stressed and thapsigargin-treated nPSCs. We compared the smOOPs mRNAs in stressed and non-stressed conditions within the same batch and observed an increased number of smOOPs transcripts under stress. This expansion of the smOOPs class upon stress supports the view that many transcripts could become increasingly condensation-prone, and as such they are extracted to a lesser extent using the standard TRIzol RNA extraction protocol.

As shown in Figure R111,J, both conditions yielded significant enrichment for SG-resident RNAs (hypergeometric test, $-\log_{10}p > 25$). The non-stressed smOOPs fraction shows a higher fold enrichment (Observed/Expected $\approx 2.2\times$), while the stressed condition captures a larger absolute number of SG transcripts ($n = 235$ vs. $n = 175$) and reaches greater statistical significance. These results raise exciting insights about the behaviour of smOOPs under stress and indicate that smOOPs recover RNAs associated with RNP granules even in the absence of stress, and this capacity increases further during granule formation.

These results support the notion that smOOPs can dynamically respond to cellular context and redistribute into defined condensates under stress.

Figure R11: A) Mapping of smOOPs onto known RNP granules/membraneless compartments. B) Enrichment of RNAs within characterised RNP granules in the smOOPs set. Overrepresentation was assessed using a hypergeometric test, using only granules data with at least 10 genes not overlapping other granules. C) Enrichment of all identified smOOPs and non-smOOPs in P-bodies relative to total RNA (log₂FC), as determined by (Hubstenberger et al. 2017). D) Representative HCR-FISH photomicrographs coupled with DDX6 immunofluorescence staining (scale bar: 5 μ m). E) Quantification of HCR-FISH coupled with DDX6 IF staining showing the percentage of RNA signal colocalising with P-bodies for each selected transcript (the number of histograms depicts the number of frames used for quantification, total number of cells counted 50-70). F) Enrichment of all identified smOOPs and non-smOOPs in stress

granules relative to total RNA (log₂FC), as determined by (Khong et al. 2017). G) Representative HCR-FISH photomicrographs coupled with G3BP1 immunofluorescence staining (scale bar: 10 μ m). H) Quantification of HCR-FISH coupled with G3BP1 IF staining showing the average percentage of RNA signal colocalising with stress granules for the selected smOOPs (n=11) and non-smOOPs (n=5). I) Overlap between smOOPs identified in non-stressed and thapsigargin-treated nPSCs. J) Enrichment of SG-resident RNAs (Khong et al. 2017) in the smOOPs pool in each condition.

While we acknowledge that direct functional perturbation experiments, such as knockdown or overexpression, would provide evidence for biological relevance, these efforts are ongoing and will require carefully designed readouts of condensate integrity without impacting the cellular translational output and downstream transcriptome effects. Because most smOOPs encode proteins, dissecting their contribution to intracellular RNA organisation without disrupting their translation poses a substantial experimental challenge. Moreover, such perturbations demand well-defined hypotheses: do smOOPs act as early scaffolds or are they recruited after granule formation? Which types of granules are involved? Which RNA regions and RBPs mediate these interactions? Our computationally driven study provides a pioneering framework for identifying and characterizing smOOPs, and our additional analyses reinforce their link to known condensates and relevance to stress. We agree that determining whether smOOPs actively contribute to condensate formation is a key next step, which we will pursue in follow-up studies aimed at addressing these mechanistic questions in depth.

2. In Figure 1D, the x-axis label for 'protein-coding' appears to be misaligned. Please adjust the label to its correct position for clarity.

We thank the reviewer for pointing out the misalignment of the x-axis label in Figure 1D, which has now been corrected.

Reviewer #3: Comments enter in this field will be shared with the author; your identity will remain anonymous.

In this manuscript Klobučar and colleagues describe a transcriptomics assay to identify condensation-prone RNAs (smOOPs). These transcripts localize to large intracellular foci, form denser RNA-RNA interaction subnetworks than expected, and are extensively bound by RNA-binding proteins (RBPs). Through an explainable DL framework, the authors uncover that smOOPs possess distinct sequence features, including reduced complexity, enhanced intramolecular folding, and specific RBP binding patterns. Notably, these RNAs encode proteins with extensive intrinsically disordered regions and are strongly predicted to participate in biomolecular condensates, highlighting the interplay between RNA- and protein-based features in phase separation. This study is quite intriguing and certainly of general interest. It advances our understanding of condensation-prone RNAs and provides a valuable resource for exploring RNA-driven condensation mechanisms. Before this work should be considered for publication a few issues should be addressed. Code and sequencing information are well documented.

1. The authors argue that their findings suggest that smOOPs are a unique class of semi-extractable transcripts that are highly bound by RBPs and form larger foci within cells expression level smOOPS RNAs versus extraction. Are these foci distinct from nuclear and cytosolic bodies or smaller and occur all over respective compartment? Co-staining with specific cellular marker would help to explain this issue.

We thank the reviewer for raising this important point regarding the spatial characteristics of the observed smOOPs foci. To address this, we performed HCR-FISH in combination with immunofluorescence staining against markers of known condensates, in particular P-bodies (DDX6) and stress granules (G3BP1). Our imaging data suggest that individual smOOPs foci colocalise to some extent with P-bodies and stress granules, with a maximum of ~40% of the RNA signal colocalising with known condensates (Figure R12A-D). Our analysis reveals that smOOPs foci exhibit a diameter of approximately 300-400 nm, which is similar to the diameter of P-bodies (determined by staining against DDX6). In contrast, the diameter of stress granules is almost five times larger (median diameter 1.4 μm). While the size of P-bodies can vary quite a lot between organisms (reviewed in (Aizer et al. 2014)), our observation is consistent with previous literature showing that cytosolic condensates such as P-bodies and stress granules typically exist on the submicron to micron scale (Freedman 2020; Visser, Lipiński, and Spruijt 2024).

Figure R12: Colocalisation of smOOPs with stress granules and P-bodies. A) Representative HCR-FISH photomicrographs (scale bar: 5 μ m). B) Quantification of HCR-FISH coupled with DDX6 IF staining showing the percentage of RNA signal colocalising with P-bodies for each selected transcript (the number of histograms depicts the number of frames used for quantification, total number of cells counted 50-70). C) Representative HCR-FISH photomicrographs (scale bar: 10 μ m). D) Quantification of HCR-FISH coupled with G3BP1 IF staining showing the percentage of RNA signal colocalising with stress granules for the selected smOOPs (n=11) and non-smOOPs (n=5). For quantification, only cells with stress granules were selected (that was between 18 and 40 cells for each transcript). The blue dots on the violin plot represent a proportion determined per image frame acquired and the black dots represent the mean. E) Enrichment of RNAs within characterised RNP granules in the smOOPs set. Overrepresentation was assessed using a hypergeometric test, using only granules data with at least 10 genes not overlapping other granules.

Moreover, we calculated the enrichment of smOOPs within known RNP granules which we now include in the **revised Figure 1C,D**. While smOOPs appear to be enriched in P-bodies (Hubstenberger et al. 2017) and density-based extracted granule fraction

(Villanueva et al. 2024), the strongest enrichment was observed in stress granules (Khong et al. 2017) (Figure R12E and revised Figure 1C,D).

2. Are mRNA transcripts with an encoded signal peptide and/or transmembrane domain excluded from the two identified clusters? Meaning ER-targeted transcript are not condensation-prone.

We thank the reviewer for this question, which touches on an important aspect of mRNA localisation and function. All our analyses on smOOPs were performed without filtering. However, we analysed the annotated signal peptides and transmembrane domains (from Uniprot) for all smOOPs and non-smOOPs. The analysis revealed that the C-rich smOOPs contained the most transcripts with an encoded signal peptide (17.3 %), the non-smOOPs 11.2 %, while the A/U-rich cluster contained the fewest (5.8 %) (Figure R13A). The same trend applies to the percentage of proteins with a transmembrane domain, with C-rich smOOPs having a slightly higher percentage than non-smOOPs (24.9 % and 22.1 %, respectively), while the A/U-rich smOOPs contained the least (11.7 %) (Figure R13B).

Figure R13: A) Percentage of proteins within non-smOOPs, C-rich smOOPs and A/U-rich smOOPs containing signal peptide (as annotated in Uniprot). B) Percentage of proteins within non-smOOPs, C-rich smOOPs and A/U-rich smOOPs containing transmembrane domain (as annotated in Uniprot). C) Mean proportion of non-smOOPs, C-rich smOOPs and A/U-rich smOOPs found within TIS granules, rough ER and cytoplasm, as identified by particle sorting by (Horste et al. 2023).

To determine whether this affects mRNA localisation, we compared smOOPs with TIS granules and rough ER-enriched transcripts in HEK293T cells (Horste et al. 2023). Interestingly, C-rich smOOPs appear to be more localised in the rough ER (p-value =

4.184e-53) (Figure R13C), suggesting that a subset of smOOPs may correspond to ER-localised transcripts, where local translation of signal peptide-containing proteins is known to occur (Jan, Williams, and Weissman 2014; Xia et al. 2019). Having only a small subset of locally translated mRNAs among the smOOPs also explains why we did not observe any changes in the overall translation efficiency (revised Figure S4H). As this interpretation remains speculative without further experiments, we have not included these results in the main manuscript, apart from the enrichment of smOOPs in RNP granules (revised Figure 1C,D).

3. Is there any evidence that smOOPs lncRNAs are being translated? Analyzing mESC ribo-seq data for novel ORFs would provide an answer to this question

We agree with the reviewer on the importance of investigating the potential translation of smOOPs lncRNAs. To address this, we reprocessed high-quality public ribosome profiling data from mouse embryonic stem cells (Tuck et al. 2020); accession: GSM3943973 and GSM3943975) using the riboseq-flow pipeline (Iosub, Wilkins, and Ule 2024). We then applied RiboCode (Xiao et al. 2018) to predict translated ORFs from ribosome-protected fragments (RPFs) (Figure R14F). Among the smOOPs lncRNAs, only four were predicted to contain translated ORFs. Given the similarity in features between lncRNA smOOPs and TECs, and the high prediction score of TECs as lncRNAs by lncRNAnet (Baek et al. 2018) (Figure R14E), we also included TECs in the analysis, but found no evidence of translation. These findings suggest that translation is rare or absent among smOOPs lncRNAs and TECs. Given the proportion of TECs increased the most in smOOPs among all gene biotypes we have now included the prediction of TECs as lncRNAs and ORF prediction results in the main manuscript (revised Figure S1E,F).

Figure R14 and revised Figure S1E,F: E) lncRNAnet prediction score for TECs (in all smOOPs) with 1 indicating prediction as lncRNA. F) Bar charts showing smOOPs lncRNAs and TECs translated ORF counts, obtained with RiboCode (Xiao et al. 2018).

4. In Fig4E the authors show that longer transcript are enriched? Which specific mRNA compartments show this enrichment? Long 5'UTR, CDS, or 3'UTR.

We thank the reviewer for their comment. We were unsure which specific aspect the question refers to, as it does not appear to directly relate to Figure 4E (which shows A/U sequence bias in cluster 2 smOOPs). If the reviewer is referring to the CDS and UTR length analysis, we hope the following clarification addresses the point accurately. As quantified in Supplementary Figure S3B, smOOPs-associated transcripts are significantly longer across all regions (two-sided t-tests: 5' UTR $p < 0.01$; CDS and 3' UTR $p < 0.001$). Notably, smOOPs CDSs are on average ~3-fold longer than those of background transcripts, making expanded coding sequences the primary driver of the overall length enrichment. This length bias is mirrored at the protein level: smOOPs proteins are not only longer but also enriched in intrinsically disordered regions, suggesting that smOOPs encode large, disordered proteins.

5. There are a number of RNA occupancy profiles published in different cell lines. Do the smOOPs transcript also show increased RBP binding also in other cell types?

We thank the reviewer for this important question. To assess whether smOOPs transcripts also exhibit elevated RBP occupancy beyond our system, we explored the global PAR-CLIP dataset in HEK293 cells (Baltz et al. 2012). This resource offers transcriptome-wide nucleotide-resolution maps of RNA-protein contacts by metabolic labeling with 4-thiouridine and 6-thioguanosine (4SU/6SG). Importantly, they prepared four matched libraries: total RNA from unlabelled, non-crosslinked cells; total RNA from 4SU/6SG-labeled, non-crosslinked cells; biotin-pulldown RNA from 4SU/6SG-labeled, non-crosslinked cells; and total RNA from UV-crosslinked, 4SU/6SG-labeled cells, thus enabling rigorous normalization.

When we normalize the PAR-CLIP signal to the transcript abundance of the biotin-pulldown library (which corrects for 4SU/6SG incorporation efficiency), smOOPs transcripts (both clusters) display subtle, yet significantly higher RBP binding than non-smOOPs (Figure R15A). In contrast, examining only the raw UV-crosslinked RNA abundance reveals a drop in smOOPs recovery, analogous to orthogonal organic phase separation (OOPS) effect in which heavily protein-bound RNAs are sequestered into the interphase during phenol-chloroform extraction and are thus missing in the aqueous fraction (Figure R15B). Together, these findings highlight an example that - while to a lesser degree - the smOOPs RNAs are more protein bound even in other systems.

Figure R15: PAR-CLIP data in 4SU/6SG-labeled HEK293 cells from (Baltz et al. 2012). A) Normalized UV-crosslinking-induced T→C transition frequencies (corrected by non-crosslinked biotin-pulldown) for

smOOPs versus non-smOOPs. B) RNA abundance ($\log_2(\text{FPKM})$) from the aqueous phase in UV-crosslinked samples, reflecting the depletion of protein-bound RNA from the aqueous phase (due to transition in the interphase) for smOOPs versus non-smOOPs.

6. The RNA occupancy profile of cluster 1 is biased, since cluster 1 transcripts are mostly composed of mRNAs. Elongating ribosomes likely displace RBPs and reduce RBP-RNA interactions. Are these CDSs more bound by RBPs than in non-smOOPs? This would have implication why these transcripts condensate. The authors should repeat the global iCLIP analysis in the presence of ATP depletion or cycloheximide, a strong translation repressor, to assess binding of RBPs in absence of translation. The authors found that the global iCLIP and POSTAR3 were most informative only in the 3'UTR. This might be again due to ribosome elongation and RBP displacement.

We thank the reviewer for raising this mechanistic point. To address whether CDS are more bound by RBPs in smOOPs than in non-smOOPs, we first compared RBP occupancy in CDS and 3'UTRs of smOOPs and non-smOOPs. Global iCLIP enrichment is observed in both regions, with a fold change $\log_2(\text{smOOPs}/\text{non-smOOPs})$ ratio of ~ 0.4 ($\sim 1.3\times$) for CDSs and ~ 0.8 ($\sim 1.7\times$) for 3'UTRs (Figure R16A), indicating that RBP binding is indeed more pronounced in 3'UTRs.

To better understand the interplay between translation and RBP binding across smOOPs, we agree that experiments under translation-inhibitory conditions would provide clearer insights. General ATP depletion would, to our understanding, indeed arrest translation by halting elongation, but at the same time also release many RBPs from energy-dependent remodeling cycles (including helicases), while cycloheximide would freeze elongating ribosomes in place, which may still sterically block RBP access and in turn global iCLIP could result in profiling ribosomal coverage. We hence tested alternative translation stressors, puromycin (causing ribosome drop-off and disassembly without freezing) and thapsigargin (specific inhibitor of the ER Ca²⁺-ATPase, leading to reduced ribosome occupancy (Wu et al. 2021)). mESC detached completely following puromycin treatment (1 μ g/ml, 2.5 hours), therefore we performed further experiments with thapsigargin treatment (10 μ M, 30 min) (Figure R12C,D) and performed OOPS and semi-extractability RNA-seq experiments with and without thapsigargin pulse treatment.

OOPS is specifically designed to isolate RNAs with increased RBP binding and thus to detect transcripts that are bound in RNP complexes. We compared OOPS-enriched RNAs in thapsigargin-treated and untreated conditions within the same batch and observed an increased number of OOPS-enriched RNAs after 30 minutes of thapsigargin treatment (Figure R16B). This expansion of the OOPS class under stress supports the

view that many transcripts may become increasingly susceptible to condensation upon translational inhibition - however, we cannot establish whether this observation is caused by translation repression or whether it is induced by SG formation or other stress-triggered pathways. It is important to note that even such a short thapsigargin pulse (10 μ M, 30 min) induced the formation of visible stress granules and increased the number of smOOPs RNAs (Figure R16C), consistent with the stress-induced RNA condensation described previously (Glauninger et al. 2024; Ren et al. 2023; Parker, Tauber, and Parker 2025; Helton, Dodd, and Moon 2025), altogether showing that translational repression is a strong perturbation and is outside the homeostatic conditions that are the focus of this study.

Figure R16: A) Length and expression normalised global iCLIP counts ratio between nPSCs smOOPs and non-smOOPs for the coding sequence and 3'UTR. B) Overlap between semi-extractable and OOPS-enriched transcripts, identified in non-stressed and thapsigargin-treated nPSCs. C) Overlap between smOOPs identified in non-stressed and thapsigargin-treated nPSCs.

To properly address this issue, we believe a time-resolved series combining translational repression, stress granule monitoring and smOOPs profiling (semi-extractability + OOPS) and ribosome profiling would be required to distinguish translation-dependent RBP binding from granule-induced condensation. We plan to investigate this systematically in follow-up work; such experiments are beyond the scope of this manuscript, which is focused on RNA condensation under developmental, non-stressful conditions. We also note that our thapsigargin experiment, while supporting a role for ribosome disengagement, induces stress granules, making it difficult to separate translational effects from SG-driven condensation. This would be problematic even if alternative translation inhibitors such as ATP depletion were used, which also induce SG formation (Wang et al. 2022).

Taken together, our iCLIP data analyses and the results of the thapsigargin experiment do not rule out that ribosome occupancy can restrict RBP binding in smOOPs mRNAs. However, not all smOOPs behave in this way: one of our top homotypically clustering smOOPs, *Dync1h1*, is predicted to condense even during active translation, raising the possibility that some RNAs form foci at sites of ongoing translation (Pichon et al. 2016). This emphasises that the interplay between translation, RBP binding and condensation is complex and cannot be fully resolved from transcriptomic data alone.

6. The authors should cross-reference their smOOPs transcript, separated in cluster 1 and 2 transcripts, with various datasets (e.g PMID: 24582499, 35020268). In Fig S4G+H, they use single ribo-seq and RNA half-life datasets to compare to all smOOPs. What about cluster 1 and 2? How do other mESC ribo-seq and half-life data relate? TE might not a good measure, since ribosome stalling is not captured and could result in increased TE. Does ribosome stalling distinguish smOOPs from non-smOOPs transcripts? There are data now on mRNA cellular flow (PMID: 38753883, 38964322, 39548324)? Are there any differences detectable for kinetic parameters of smOOPs vs non-smOOPs transcripts?

We thank the reviewer for these thoughtful suggestions. In response, we performed separate analyses for clusters 1 and 2 of smOOPs transcripts to examine transcript stability, translational dynamics, and subcellular RNA kinetics in greater detail.

To explore whether translation efficiency (TE) in smOOPs transcripts reflects true increases in productive translation or rather differences in ribosome dynamics, we analysed additional publicly available ribosome profiling data (Tuck et al. 2020). This dataset provides TE estimates separately derived from monosome and disome footprints, which respectively reflect individual ribosome occupancy and ribosome collisions. In wild-type cells, we observed that nPSCs smOOPs transcripts, particularly the A/U-rich cluster, display elevated monosome- and disome-derived TE relative to non-smOOPs transcripts (Figure R17A). This indicates that smOOPs transcripts tend to be more frequently loaded with ribosomes and exhibit a higher rate of ribosome collisions. Although monosome data shows a significant effect that was not seen in the TE calculations of (Ingolia, Lareau, and Weissman 2011), we have retained it in the manuscript (revised Figure S4H) as it includes TE measurements for more than three times as many genes.

We reassessed transcript stability (Steinbrecht et al. 2024) for each cluster of nPSCs smOOPs individually. Cluster 1 transcripts show a modest reduction in overall stability compared to controls, whereas cluster 2 transcripts appear largely unchanged (Figure R17B). We further interrogated the dataset which applies a cellular RNA flow framework to model RNA stability across different stages of the RNA life cycle; including nuclear

pre-mRNA, mature nuclear RNA, cytosolic RNA, and membrane-associated RNA fractions (Steinbrecht et al. 2024). We observed a significant decrease in half-life in the pre-mRNA processing and cytosol stage of the lifecycle for cluster 1, with an increase in membrane half-life for both clusters. Interestingly, we also observed a decrease in whole-cell half-life but not in the model derived half-life (Figure R17B). Next, we investigated the kinetic parameters such as nuclear export rates and cytoplasmic decay (Steinbrecht et al. 2024). Cluster 1 smOOPs show slower nuclear decay and faster pre-mRNA processing with accelerated cytosolic removal relative to controls (Figure R17C). Importantly, both clusters exhibit slower membrane decay with cluster 1 showing higher significance. These findings support the hypothesis that smOOPs transcripts might be processed differently depending on cluster identity, with implications for their function and role. Due to Cell Press space limitations requiring an equal number of main and supplementary figures, we propose including this content in the transparent revision file instead.

Figure R17: A) Translation efficiency derived from monosome (left) and disome (right) profiling (Tuck et al. 2020), comparing non-smOOPs to nPSCs smOOPS transcripts in clusters 1 and 2. B) RNA half-lives (min) measured at successive stages of the mRNA life cycle: pre-mRNA processing; nuclear retention; cytosolic stability; membrane stability; and whole-cell decay (direct measurement and model-derived), comparing Non-smOOPs to nPSC smOOPS transcripts in clusters 1 and 2. C) RNA processing rates (min^{-1}) for each step of the mRNA life cycle: pre-mRNA processing; nuclear export; nuclear decay; nuclear removal (export + decay); cytosolic transport; cytosolic decay; cytosolic removal (transport + decay); and membrane decay, comparing non-smOOPs to nPSC smOOPS transcripts in clusters 1 and 2.

7. The authors describe, the ratio of C-rich to A/U-rich smOOPs shifts during development. In nPSCs, C-rich smOOPs were predominant, comprising approximately 75% of nPSCs-specific smOOPs. In dPSCs, their proportion declined to 56%, reflecting an increased representation of A/U-rich smOOPs at later developmental stages. Why is this the case? Does the cell cycle length or cell cycle distribution at different stages play a role? Past work showed that tRNA expression profile in proliferating and differentiated cells influences codon biases (PMID: 25215487). Could this have an impact here?

We thank the reviewer for this intriguing question. We believe that the developmental shift from C-rich to A/U-rich smOOPs is likely multifactorial and at this point as such speculative. Besides the codon usage biases shaped by tRNA availability (Gingold et al. 2014), it is worth noting that the control of pluripotency progression is very dynamic and multi-layered (Yang et al. 2019), with significant RBPome changes across development (Modic, de Los Mozos, and Steinhauser 2022; Kastelic 2021).

References

- Aizer, Adva, Alon Kalo, Pinhas Kafri, Amit Shraga, Rakefet Ben-Yishay, Avi Jacob, Noa Kinor, and Yaron Shav-Tal. 2014. "Quantifying mRNA Targeting to P-Bodies in Living Human Cells Reveals Their Dual Role in mRNA Decay and Storage." *Journal of Cell Science* 127 (Pt 20): 4443–56.
- Arimoto-Matsuzaki, Kyoko, Haruo Saito, and Mutsuhiro Takekawa. 2016. "TIA1 Oxidation Inhibits Stress Granule Assembly and Sensitizes Cells to Stress-Induced Apoptosis." *Nature Communications* 7 (1): 10252.
- Baek, Junghwan, Byunghan Lee, Sunyoung Kwon, and Sungroh Yoon. 2018. "LncRNA-net: Long Non-Coding RNA Identification Using Deep Learning." *Bioinformatics (Oxford, England)* 34 (22): 3889–97.
- Baltz, Alexander G., Mathias Munschauer, Björn Schwanhäusser, Alexandra Vasile, Yasuhiro Murakawa, Markus Schueler, Noah Youngs, et al. 2012. "The mRNA-Bound Proteome and Its Global Occupancy Profile on Protein-Coding Transcripts." *Molecular Cell* 46 (5): 674–90.
- Cai, Zhaokui, Changchang Cao, Lei Ji, Rong Ye, Di Wang, Cong Xia, Sui Wang, et al. 2020. "RIC-Seq for Global in Situ Profiling of RNA–RNA Spatial Interactions." *Nature* 582 (7812): 432–37.
- Choi, Harry M. T., Maayan Schwarzkopf, Mark E. Fornace, Aneesh Acharya, Georgios Artavanis, Johannes Stegmaier, Alexandre Cunha, and Niles A. Pierce. 2018. "Third-Generation in Situ Hybridization Chain Reaction: Multiplexed, Quantitative, Sensitive, Versatile, Robust." *Development* 145 (12). <https://doi.org/10.1242/dev.165753>.
- Chouaib, Racha, Adham Safieddine, Xavier Pichon, Arthur Imbert, Oh Sung Kwon, Aubin Samacoits, Abdel-Meneem Traboulsi, et al. 2020. "A Dual Protein-mRNA Localization Screen Reveals Compartmentalized Translation and Widespread Co-Translational RNA Targeting." *Developmental Cell* 54 (6): 773–91.e5.
- Demeshkina, Natalia A., and Adrian R. Ferré-D'Amaré. 2025. "Large-Scale Purifications Reveal Yeast and Human Stress Granule Cores Are Heterogeneous Particles with Complex Transcriptomes and Proteomes." *Cell Reports* 44 (6): 115738.
- Fazal, Furqan M., Shuo Han, Kevin R. Parker, Pornchai Kaewsapsak, Jin Xu, Alistair N. Boettiger, Howard Y. Chang, and Alice Y. Ting. 2019. "Atlas of Subcellular RNA Localization Revealed by APEX-Seq." *Cell* 178 (2): 473–90.e26.
- Freedman, Miriam Arak. 2020. "Liquid-Liquid Phase Separation in Supramicrometer and Submicrometer Aerosol Particles." *Accounts of Chemical Research* 53 (6): 1102–10.
- Gingold, Hila, Disa Tehler, Nanna R. Christoffersen, Morten M. Nielsen, Fazila Asmar, Susanne M. Kooistra, Nicolaj S. Christoffersen, et al. 2014. "A Dual Program for Translation Regulation in

- Cellular Proliferation and Differentiation." *Cell* 158 (6): 1281–92.
- Glauninger, Hendrik, Jared A. M. Bard, Caitlin J. Wong Hickernell, Edo M. Airoidi, Weihan Li, Robert H. Singer, Sneha Paul, et al. 2024. "Transcriptome-Wide mRNA Condensation Precedes Stress Granule Formation and Excludes Stress-Induced Transcripts." *bioRxiv*.
<https://doi.org/10.1101/2024.04.15.589678>.
- Helton, Noah S., Benjamin Dodd, and Stephanie L. Moon. 2025. "Ribosome Association Inhibits Stress-Induced Gene mRNA Localization to Stress Granules." *Genes & Development* 39 (13-14): 826–48.
- Horste, Ellen L., Mervin M. Fansler, Ting Cai, Xiuzhen Chen, Sibylle Mitschka, Gang Zhen, Flora C. Y. Lee, Jernej Ule, and Christine Mayr. 2023. "Subcytoplasmic Location of Translation Controls Protein Output." *Molecular Cell* 83 (24): 4509–23.e11.
- Hubstenberger, Arnaud, Maité Courel, Marianne Bénard, Sylvie Souquere, Michèle Ernoult-Lange, Racha Chouaib, Zhou Yi, et al. 2017. "P-Body Purification Reveals the Condensation of Repressed mRNA Regulons." *Molecular Cell* 68 (1): 144–57.e5.
- Ingolia, Nicholas T., Liana F. Lareau, and Jonathan S. Weissman. 2011. "Ribosome Profiling of Mouse Embryonic Stem Cells Reveals the Complexity and Dynamics of Mammalian Proteomes." *Cell* 147 (4): 789–802.
- Iosub, Ira A., Oscar G. Wilkins, and Jernej Ule. 2024. "Riboseq-Flow: A Streamlined, Reliable Pipeline for Ribosome Profiling Data Analysis and Quality Control." *Wellcome Open Research* 9 (April):179.
- Jan, Calvin H., Christopher C. Williams, and Jonathan S. Weissman. 2014. "Principles of ER Cotranslational Translocation Revealed by Proximity-Specific Ribosome Profiling." *Science (New York, N.Y.)* 346 (6210): 1257521.
- Kastelic, Nicolai. 2021. "RNA-Based Regulation of Pluripotency and Differentiation." <https://core.ac.uk/reader/541480742>.
- Khong, Anthony, Tyler Matheny, Saumya Jain, Sarah F. Mitchell, Joshua R. Wheeler, and Roy Parker. 2017. "The Stress Granule Transcriptome Reveals Principles of mRNA Accumulation in Stress Granules." *Molecular Cell* 68 (4): 808–20.e5.
- Modic, M., I. R. de Los Mozos, and S. Steinhauser. 2022. "Epiblast Morphogenesis Is Controlled by Selective mRNA Decay Triggered by LIN28A Relocation." *bioRxiv*.
<https://www.biorxiv.org/content/10.1101/2021.03.15.433780v1.abstract>.
- Parker, Dylan M., Devin Tauber, and Roy Parker. 2025. "G3BP1 Promotes Intermolecular RNA-RNA Interactions during RNA Condensation." *Molecular Cell* 85 (3): 571–84.e7.
- Pattabiraman, Sundararaghavan, Gajendra Kumar Azad, Triana Amen, Shlomi Brielle, Jung Eun Park, Siu Kwan Sze, Eran Meshorer, and Daniel Kaganovich. 2020. "Vimentin Protects Differentiating Stem Cells from Stress." *Scientific Reports* 10 (1): 19525.
- Pichon, Xavier, Amandine Bastide, Adham Safieddine, Racha Chouaib, Aubin Samacoits, Eugenia Basyuk, Marion Peter, Florian Mueller, and Edouard Bertrand. 2016. "Visualization of Single Endogenous Polysomes Reveals the Dynamics of Translation in Live Human Cells." *The Journal of Cell Biology* 214 (6): 769–81.
- Ren, Ziqi, Wei Tang, Luxin Peng, and Peng Zou. 2023. "Profiling Stress-Triggered RNA Condensation with Photocatalytic Proximity Labeling." *Nature Communications* 14 (1): 7390.
- Ries, Ryan J., Sara Zaccara, Pierre Klein, Anthony Olarerin-George, Sim Namkoong, Brian F. Pickering, Deepak P. Patil, Hojoong Kwak, Jun Hee Lee, and Samie R. Jaffrey. 2019. "m6A Enhances the Phase Separation Potential of mRNA." *Nature* 571 (7765): 424–28.
- Steinbrecht, David, Igor Minia, Miha Milek, Johannes Meisig, Nils Blüthgen, and Markus Landthaler. 2024. "Subcellular mRNA Kinetic Modeling Reveals Nuclear Retention as Rate-Limiting." *Molecular Systems Biology* 20 (12): 1346–71.
- Trcek, Tatjana, Tyler E. Douglas, Markus Grosch, Yandong Yin, Whitby V. I. Eagle, Elizabeth R. Gavis, Hari Shroff, Eli Rothenberg, and Ruth Lehmann. 2020. "Sequence-Independent Self-Assembly of Germ Granule mRNAs into Homotypic Clusters." *Molecular Cell* 78 (5): 941–50.e12.
- Tuck, Alex Charles, Aneliya Rankova, Alaaddin Bulak Arpat, Luz Angelica Liechti, Daniel Hess, Vytautas Iesmantavicius, Violeta Castelo-Szekely, David Gatfield, and Marc Bühler. 2020. "Mammalian RNA Decay Pathways Are Highly Specialized and Widely Linked to Translation." *Molecular Cell* 77 (6): 1222–36.e13.
- Villanueva, Eneko, Tom Smith, Mariavittoria Pizzinga, Mohamed Elzek, Rayner M. L. Queiroz,

Robert F. Harvey, Lisa M. Breckels, et al. 2024. "System-Wide Analysis of RNA and Protein Subcellular Localization Dynamics." *Nature Methods*, January. <https://doi.org/10.1038/s41592-023-02101-9>.

Visser, Brent S., Wojciech P. Lipiński, and Evan Spruijt. 2024. "The Role of Biomolecular Condensates in Protein Aggregation." *Nature Reviews Chemistry* 8 (9): 686–700.

Wang, Tao, Xibin Tian, Han Byeol Kim, Yura Jang, Zhiyuan Huang, Chan Hyun Na, and Jiou Wang. 2022. "Intracellular Energy Controls Dynamics of Stress-Induced Ribonucleoprotein Granules." *Nature Communications* 13 (1): 5584.

Wu, I-Hui, Jae Seok Yoon, Qian Yang, Yi Liu, William Skach, and Philip Thomas. 2021. "A Role for the Ribosome-Associated Complex in Activation of the IRE1 Branch of UPR." *Cell Reports* 36 (2): 109366.

Xia, Chenglong, Jean Fan, George Emanuel, Junjie Hao, and Xiaowei Zhuang. 2019. "Spatial Transcriptome Profiling by MERFISH Reveals Subcellular RNA Compartmentalization and Cell Cycle-Dependent Gene Expression." *Proceedings of the National Academy of Sciences of the United States of America* 116 (39): 19490–99.

Xiao, Zhengtao, Rongyao Huang, Xudong Xing, Yuling Chen, Haiteng Deng, and Xuerui Yang. 2018. "De Novo Annotation and Characterization of the Translatome with Ribosome Profiling Data." *Nucleic Acids Research* 46 (10): e61.

Yang, Pengyi, Sean J. Humphrey, Senthilkumar Cinghu, Rajneesh Pathania, Andrew J. Oldfield, Dharendra Kumar, Dinuka Perera, et al. 2019. "Multi-Omic Profiling Reveals Dynamics of the Phased Progression of Pluripotency." *Cell Systems*. <https://doi.org/10.1016/j.cels.2019.03.012>.

Zeng, Chao, Takeshi Chujo, Tetsuro Hirose, and Michiaki Hamada. 2023. "Landscape of Semi-Extractable RNAs across Five Human Cell Lines." *Nucleic Acids Research* 51 (15): 7820–31.

Zhou, Yilong, Amol Panhale, Maria Shvedunova, Mirela Balan, Alejandro Gomez-Auli, Herbert Holz, Janine Seyfferth, et al. 2024. "RNA Damage Compartmentalization by DHX9 Stress Granules." *Cell* 187 (7): 1701–18.e28.

Referees' report, second round of review

Reviewer 1:

The authors have adequately addressed my comments and I am supportive of publication.

A few minor errors to correct:

1) I couldn't figure out what none vs novel meant for figure S1F so a definition would be helpful there.

2) The microscopy labels are messed up in 11

3) The second panel in S5F has "vatio" instead of "ratio"

Reviewer 2:

The new experiments, additional analyses, and expanded methodological details substantially enhance the reproducibility and interpretability of the manuscript. Overall, the major concerns have been adequately addressed.

Reviewer 3:

The authors performed a thorough revision and have addressed the concerns and suggestions that I raised during the first round of review.

Authors' response to the second round of review